# BiasScope: Towards Automated Detection of Bias in LLM-as-a-Judge Evaluation

**Peng Lai**[1][*] **Zhihao Ou**[1][*] **Yong Wang**[2]**, Longyue Wang**[2]**, Jian Yang**[3]**, Yun Chen**[4]
**Guanhua Chen**[1][†]
[1]Southern University of Science and Technology, [2]Alibaba Group
[3]Beihang University, [4]Shanghai University of Finance and Economics

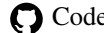 Code 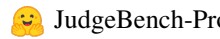 JudgeBench-Pro

## Abstract

LLM-as-a-Judge has been widely adopted across various research and practical applications, yet the robustness and reliability of its evaluation remain a critical issue. A core challenge it faces is bias, which has primarily been studied in terms of known biases and their impact on evaluation outcomes, while automated and systematic exploration of potential unknown biases is still lacking. Nevertheless, such exploration is crucial for enhancing the robustness and reliability of evaluations. To bridge this gap, we propose BiasScope, a LLM-driven framework for automatically and at scale discovering potential biases that may arise during model evaluation. BiasScope can uncover potential biases across different model families and scales, with its generality and effectiveness validated on the JudgeBench dataset. It overcomes the limitations of existing approaches, transforming bias discovery from a passive process relying on manual effort and predefined bias lists into an active and comprehensive automated exploration. Moreover, based on BiasScope, we propose JudgeBench-Pro, an extended version of JudgeBench and a more challenging benchmark for evaluating the robustness of LLM-as-a-judge. Strikingly, even powerful LLMs as evaluators show error rates above 50% on JudgeBench-Pro, underscoring the urgent need to strengthen evaluation robustness and to mitigate potential biases further.

## 1 Introduction

With the optimization of algorithms and model architectures, the field of AI has gradually entered the second phase—the era of evaluation (Fei et al., 2025). Model improvement no longer relies solely on training; rather, it increasingly depends on practical evaluation to uncover potential shortcomings and guide further enhancement (Gu et al., 2025). LLM-as-a-Judge (Zheng et al., 2023), as a promising new paradigm, offers advantages over traditional methods by leveraging the large language model (LLM) as a "judge" to evaluate model outputs at scale in diverse and dynamic settings with automation and consistency (Wei et al., 2025; Li et al., 2025a). Moreover, LLM-as-a-Judge has now been extensively adopted across a wide range of research and application domains, including benchmark construction (Lambert et al., 2024; Tan et al., 2025), data curation (Wu et al., 2024; Chen et al., 2024b), and model performance evaluation (Zheng et al., 2023; Li et al., 2023). Consequently, given its widespread adoption, ensuring the reliability and robustness of LLM-as-a-judge has become a critical challenge that urgently needs to be addressed.

The core challenge faced by LLM-as-a-judge primarily stems from bias (Chen et al., 2024a). Bias refers to the systematic, non-random tendencies exhibited by a Judge LLM during answer evaluation, which can lead its assessments to deviate from objective and equitable standards, thereby affecting the robustness and reliability of the evaluation (Wang et al., 2023). Early studies primarily focused on verifying whether LLMs maintain robustness when affected by biases, or on mitigating

---

[*]Equal contribution.
[†]Corresponding author.

the impacts of such biases, with common types including gender bias (Prabhune et al., 2025), length bias (Ye et al., 2024), self-bias (Xu et al., 2024), position bias (Li et al., 2024), and so on. Meanwhile, related work (e.g., CALM (Ye et al., 2024)) has attempted to construct benchmarks using known biases to quantify the extent of bias exhibited by LLM-as-a-judge. However, these studies are primarily limited to verifying and analyzing known biases, lacking systematic exploration of potential or unidentified biases, which may have a more significant impact on the reliability of LLM-as-a-Judge and the fairness of its assessment outcomes. Identifying such potential biases manually is challenging to scale, which naturally raises the question: how can potential biases be discovered in an automated and large-scale manner?

To address this question, we propose BIASSCOPE, a framework that iteratively and automatically discovers potential diversity biases in the LLM evaluation process. BIASSCOPE consists of two phases: (1) Bias Discovery, a teacher model is leveraged to inject basic biases into the target dataset to trigger and identify potential biases in the target model; (2) Bias Validation, the effectiveness of candidate biases in perturbing the target model is assessed on a test dataset, and the biases confirmed to be effective are then integrated into the basic bias library. This process is then iterated to obtain more diverse and effective biases in target models continuously. We conduct reliability validation of BIASSCOPE, confirming that its observed effects are not caused by perturbations that increase response length or modify answers, and we find that incorporating preference data synthesized from the discovered biases into DPO (Rafailov et al., 2024) training further mitigates the biases exhibited by the model during evaluation.

Moreover, building upon JudgeBench (Tan et al., 2025), we use BIASSCOPE to construct a more challenging benchmark, JudgeBench-Pro, designed to evaluate the assessment capabilities and robustness of LLM-as-a-judge. This Benchmark was carefully curated through verification by powerful LLMs and rigorous manual review. The evaluation results show that, among the five mainstream powerful models, four performed at or below the level of random guessing, with an average error rate 25.9% higher than on JudgeBench. These findings indicate that ensuring the robustness of current LLM-as-a-Judge remains challenging.

To summarize, our main contributions are as follows:

▷ We propose BIASSCOPE, a framework entirely driven by large language models that can automatically and at scale discover potential biases that may arise during model evaluation.

▷ BIASSCOPE can mine potential biases in models across different families and scales, and its generality and effectiveness are validated on the objective and reliable JudgeBench dataset.

▷ Leveraging our framework BIASSCOPE, we developed JudgeBench-Pro, a more challenging benchmark for evaluating the robustness of LLMs as judges, extending the original JudgeBench.

## 2 BIASSCOPE

To systematically uncover potential biases in the target model, we propose BIASSCOPE, an iterative framework (Figure 1). The detailed pseudocode is provided in Algorithm 1. BIASSCOPE leverages random bias perturbations combined with the target model's misjudgment self-explanations to induce the model to expose more diverse potential biases, which are then analyzed and identified using a teacher model (§2.2). These biases are subsequently compared against a known bias library and validated through perturbation tests, retaining only those that are both novel and genuinely reflected in the model's behavior, thereby enabling the bias space to self-expand and self-converge (§2.3).

### 2.1 GENERAL PROBLEM FORMULATION OF AUTOMATIC BIAS DISCOVERY

In this section, we formalize the problem of automatic bias discovery in the LLM-as-a-judge paradigm. Following previous work (Tan et al., 2025; Ye et al., 2024), we adopt the pair-wise evaluation approach to identify the potential biases of LLM-as-a-Judge better and reduce confounding effects. Let $\mathcal{D} = \{(x_i, y_i^c, y_i^r)\}_{i=1}^N$ denote a target preference dataset, where $x_i$ is the input instruction, $y_i^c$ is the chosen response, and $y_i^r$ is the rejected response. We denote the target model as $M$, which is the model whose potential biases we aim to analyze. Let $\mathcal{B}_0 = \{b_1, b_2, \ldots, b_K\}$ denote the initial bias library, where each $b_k$ represents a known bias (e.g., tends to favor longer responses).

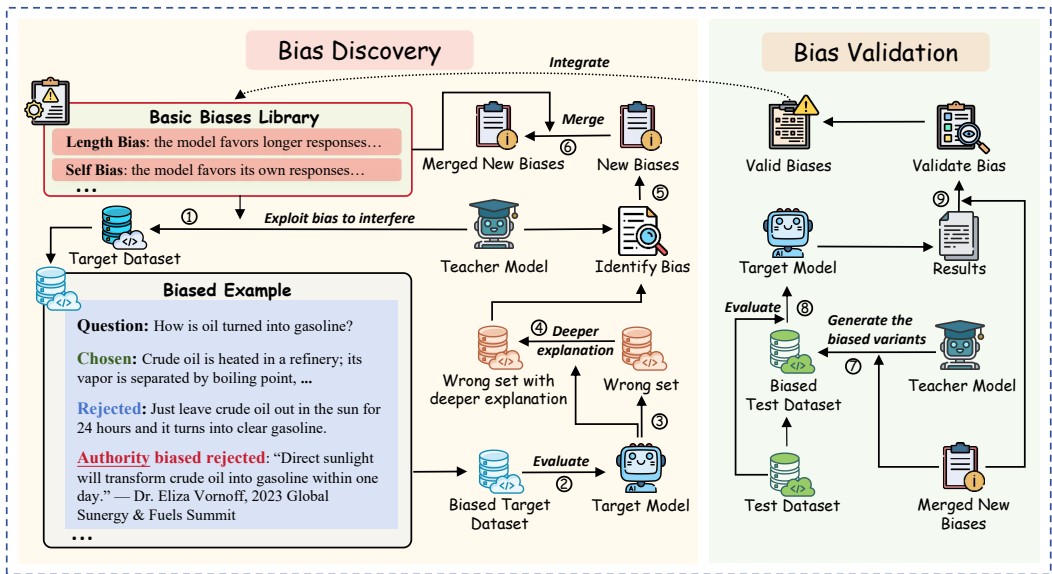

Figure 1: **The Overview of** BIASSCOPE. In the Bias Discovery phase (**Left**), we evaluate the target model on the target dataset perturbed by known biases to expose further potential biases, which are then discovered by a teacher model. In the Bias Validation phase (**Right**), we introduce a test dataset to examine the effectiveness of the discovered biases. Based on the evaluation results, valid biases are retained and incorporated into the basic bias library to support subsequent iterations.

The goal is to iteratively expand this library through two phases: discovering potential biases and validating their significance. Assume that at iteration $t$, the bias library is $\mathcal{B}_t$:

▷ In the **discovery** phase, candidate biases are generated via a function DiscoverBias($\cdot$), which systematically detects potential biases based on model outputs, explanations, or other auxiliary information $\mathcal{A}_t$. The candidate bias set $\mathcal{C}_t = \{b_{t,1}, b_{t,2}, \ldots, b_{t,M_t}\}$ is generated as

$$\mathcal{C}_t = \text{DiscoverBias}(M, \mathcal{D}, \mathcal{B}_t, \mathcal{A}_t). \tag{1}$$

▷ In the **validation** phase, each candidate bias $b \in \mathcal{C}_t$ is evaluated using a verification function Verify($b$) $\in \{0, 1\}$, which assesses the bias based on criteria such as significance (impact on judgments). A bias is deemed valid if Verify($b$) = 1. The bias library is updated as

$$\mathcal{B}_{t+1} = \mathcal{B}_t \cup \{b \mid \text{Verify}(b) = 1, b \in \mathcal{C}_t\}. \tag{2}$$

The process iterates over $t = 0, 1, \ldots, T - 1$ until convergence, which occurs when no candidate biases will be verified ($\mathcal{C}_T = \emptyset$), the bias library stabilizes ($\mathcal{B}_{T+1} = \mathcal{B}_T$), or $t$ reaches the maximum iteration $T_{\max}$ ($t = T_{\max}$). Then, the process will output the final bias library $\mathcal{B}_T$.

## 2.2 EFFICIENT BIAS DISCOVERY VIA A TEACHER MODEL

To achieve more efficient and diverse discovery, we introduce a teacher model $M_T$ to assist in this process. We apply a sampled bias $b_k \sim \mathcal{B}_t$ to each rejected response $y_i^r \in \mathcal{D}$ associated with input $x_i$, and then require the teacher to generate its biased variant $\tilde{y}_i^r$ while preserving the original outcome as much as possible, constructing a perturbed dataset $\tilde{\mathcal{D}}_t$ (step ① in Figure 1):

$$\tilde{\mathcal{D}}_t = \{(x_i, y_i^c, \tilde{y}_i^r) \mid \tilde{y}_i^r = \text{Perturb}(x_i, y_i^r, b_k; M_T), b_k \sim \mathcal{B}_t, (x_i, y_i^c, y_i^r) \in \mathcal{D})\}. \tag{3}$$

The target model $M$ is evaluated on the perturbed dataset $\tilde{\mathcal{D}}_t$, generating corresponding explanation $E_i$ and predictions $\hat{y}_i$ as $\{(\hat{y}_i, E_i)\}_{i=1}^N = \text{Evaluate}(\tilde{\mathcal{D}}_t; M)\}$, where Evaluate($\cdot; M$) represents the process of evaluating $M$. Then, we extract the misjudged instances together with their associated explanation to construct a new dataset $\tilde{\mathcal{D}}_t^{\text{mis}}$ (steps ② and ③ in Figure 1):

$$\tilde{\mathcal{D}}_t^{\text{mis}} = \{(x_i, y_i^c, \tilde{y}_i^r, E_i) \mid \{(\hat{y}_i, E_i)\}_{i=1}^N = \text{Evaluate}(\tilde{\mathcal{D}}_t; M)\}, \mathbf{1}[\hat{y}_i \neq y_i^c] = 1, (x_i, y_i^c, \tilde{y}_i^r) \in \tilde{\mathcal{D}}_t\}, \tag{4}$$

where $\mathbf{1}[\hat{y}_i \neq y_i^c]$ denotes the indicator function. Although $\tilde{\mathcal{D}}_t^{\text{mis}}$ contains instances of the model's misjudgments along with erroneous explanations, these explanations alone are insufficient to fully reveal the model's evaluation biases. To further elicit the model's potential biases, we employ an **error cascading** strategy: *the model generates deeper explanations for its own erroneous reasoning, thereby inducing more profound errors.* The effectiveness of this strategy is experimentally validated in §3.3. This process will generate explanations containing more potential biases, which then replace the original $E_i$, resulting in a dataset enriched with bias-analytical information (step ④ in Figure 1):

$$\tilde{\mathcal{D}}_t^{\text{final}} = \{(x_i, y_i^c, \tilde{y}_i^r, E_i') \mid E_i' = \text{DeeperExplain}(x_i, y_i^c, \tilde{y}_i^r, E_i; M), \ (x_i, y_i^c, \tilde{y}_i^r, E_i) \in \tilde{\mathcal{D}}_t^{\text{mis}}\}. \quad (5)$$

To ensure that the subsequently obtained biases are valid and non-overlapping, we first perform bias discovery and then merge similar biases, thereby ensuring that the resulting biases are independent. Specifically, we apply the teacher model $M_T$ to discover a new set of biases $\tilde{\mathcal{B}}_t$ (step ⑤ in Figure 1):

$$\tilde{\mathcal{B}}_t = \{b_j \mid b_j = \text{IdentifyBias}(x_i, y_i^c, \tilde{y}_i^r, E_i'; M_T), (x_i, y_i^c, \tilde{y}_i^r, E_i') \in \tilde{\mathcal{D}}_t^{\text{final}}\}. \quad (6)$$

Notably, the teacher models here focus on reasoning-based analysis rather than subjective preference judgments, so potential preference leakage (Li et al., 2025b) is avoided. Next, we construct a temporary bias set $\mathcal{B}_t^{\text{temp}} = \tilde{\mathcal{B}}_t \cup \mathcal{B}_t$, and prompt the teacher model $M_T$ to perform pairwise comparisons of all biases in $\mathcal{B}_t^{\text{temp}}$ to assess their similarity, and merge them when redundancy is detected (step ⑥ in Figure 1):

$$\hat{\mathcal{B}}_t = \{b^* \mid b^* = \text{Merge}(b_i, b_j; M_T), b_i, b_j (i \neq j) \in \mathcal{B}_t^{\text{temp}}, \mathcal{B}_t^{\text{temp}} = \tilde{\mathcal{B}}_t \cup \mathcal{B}_t\}, \quad (7)$$

where $\text{Merge}(\cdot)$ denotes the entire process of comparison and merging, while keeping $\mathcal{B}_t$ unchanged. Finally, we remove the biases that already exist in the basic bias library to obtain the final candidate bias set $\mathcal{C}_t = \hat{\mathcal{B}}_t \setminus \mathcal{B}_t$.

## 2.3 VALIDATING BIAS BASED ON A TEST DATASET

We introduce a small test dataset for validation to ensure that the potential biases identified by our framework are reasonable and valid. We denote this test dataset as $\mathcal{D}^{\text{test}} = \{(x_i, y_i^c, y_i^r)\}_{i=1}^H$. Following the procedure described at the beginning of §2.2, but with the distinction that each candidate bias $b_j$ in the candidate bias set $\mathcal{C}_t$ is used to perturb the entire test dataset $\mathcal{D}^{\text{test}}$, we use the teacher model $M_T$ to generate a perturbed test dataset $\tilde{\mathcal{D}}_j^{\text{test}}$ corresponding to each bias (step ⑦ in Figure 1):

$$\tilde{\mathcal{D}}_j^{\text{test}} = \{(x_i, y_i^c, \tilde{y}_i^r) \mid \tilde{y}_i^r = \text{Perturb}(x_i, y_i^r, b_j; M_T), b_j \in \mathcal{C}_t, (x_i, y_i^c, y_i^r) \in \mathcal{D}^{\text{test}})\}. \quad (8)$$

Ye et al. (2024) points out that when the model makes a judgment on the perturbed pair-wise data and chooses the rejected response, it can be considered to exhibit the corresponding bias. Therefore, we only need to compare the target model's error rate on the perturbed dataset $\tilde{\mathcal{D}}_j^{\text{test}}$ with that on the original dataset $\mathcal{D}^{\text{test}}$: if the former is higher than the latter, the bias can be deemed effective. Therefore, we need to evaluate the target model $M$ separately on the original test dataset $\mathcal{D}^{\text{test}}$ and the perturbed dataset $\tilde{\mathcal{D}}_j^{\text{test}}$: $(\hat{y}_i, E_i)\}_{i=1}^H = \text{Evaluate}(\mathcal{D}^{\text{test}}; M), (\hat{y}_i^j, E_i^j)\}_{i=1}^H = \text{Evaluate}(\tilde{\mathcal{D}}_j^{\text{test}}; M)$. Then, we compute the error rates (**Err**) of $M$ on the two datasets, respectively (step ⑧ in Figure 1):

$$\mathbf{Err}(\mathcal{D}^{\text{test}}) = \frac{1}{H} \sum_{i=1}^H \mathbf{1}[\hat{y}_i \neq y_i^c], \quad \mathbf{Err}(\tilde{\mathcal{D}}_j^{\text{test}}) = \frac{1}{H} \sum_{i=1}^H \mathbf{1}[\hat{y}_i^j \neq y_i^c]. \quad (9)$$

Therefore, we can proceed based on the error rates to update the bias library (step ⑨ in Figure 1):

$$\mathcal{B}_{t+1} = \mathcal{B}_t \cup \{b_j \mid \mathbf{Err}(\tilde{\mathcal{D}}_j^{\text{test}}) > \mathbf{Err}(\mathcal{D}^{\text{test}}), b_j \in \mathcal{C}_t\}. \quad (10)$$

At this point, we have fully established an automated framework for bias discovery. We present two bias examples uncovered by BIASSCOPE below; more examples can be found in Appendix H.

> **Two Representative Examples of Valid Biases Uncovered through** BIASSCOPE
>
> ▷ **Novelty Bias**: Tendency to overvalue new or unusual information, perceiving it as more important or accurate than familiar information, even when novelty $\neq$ quality.

> ▷ **Exact Match Bias**: A model tends to prefer answers that exactly match the source text or reference, even if other answers are equally correct or better.

Table 1: Impact of Biases Mined by BIASSCOPE on JudgeBench Across Multiple Target Models. "Original" denotes the model's error rate on the original JudgeBench test set, while "BIASSCOPE" denotes its average error rate on the perturbed JudgeBench samples constructed based on the corresponding effective biases identified by the BiasScope framework. Note that 50% corresponds to random chance performance.

| Target Model | Type | # Validated Biases | Error Rates (%) on JudgeBench | | | | |
|---|---|---|---|---|---|---|---|
| | | | Code | Knowl. | Math | Reason. | Overall |
| Qwen2.5-1.5B-Instruct | Original | - | 54.5 | 48.8 | 38.7 | 52.2 | 48.6 |
| | BIASSCOPE | 48 | 54.1 | 54.5 | 49.3 | 52.5 | 53.1 |
| | △ | - | -0.4 | +5.7 | +10.6 | +0.3 | +4.5 |
| InternLM3-8B-Instruct | Original | - | 52.1 | 46.1 | 40.5 | 44.0 | 45.3 |
| | BIASSCOPE | 19 | 55.7 | 49.6 | 51.2 | 49.7 | 50.7 |
| | △ | - | +3.6 | +3.5 | +10.7 | +5.7 | +5.4 |
| Mistral-7B-Instruct-v0.3 | Original | - | 43.8 | 46.5 | 32.1 | 47.7 | 43.9 |
| | BIASSCOPE | 41 | 55.2 | 53.6 | 47.9 | 47.3 | 51.2 |
| | △ | - | +11.4 | +7.1 | +15.8 | -0.4 | +7.3 |
| Qwen2.5-7B-Instruct | Original | - | 49.0 | 49.0 | 27.7 | 41.6 | 43.4 |
| | BIASSCOPE | 27 | 56.3 | 51.6 | 40.4 | 43.3 | 48.1 |
| | △ | - | +7.3 | +2.6 | +12.7 | +1.7 | +4.7 |
| LLaMA-3.1-8B-Instruct | Original | - | 52.4 | 42.3 | 26.6 | 46.9 | 41.7 |
| | BIASSCOPE | 29 | 61.5 | 53.6 | 42.3 | 53.7 | 52.5 |
| | △ | - | +9.1 | +11.3 | +15.7 | +6.8 | +10.8 |
| Qwen2.5-14B-Instruct | Original | - | 41.1 | 40.9 | 30.4 | 35.6 | 37.7 |
| | BIASSCOPE | 19 | 51.8 | 49.0 | 40.3 | 49.3 | 47.8 |
| | △ | - | +10.7 | +8.1 | +9.9 | +13.7 | +10.1 |
| Qwen3-8B (Non-Tinking) | Original | - | 39.7 | 40.0 | 27.9 | 36.1 | 36.9 |
| | BIASSCOPE | 14 | 45.6 | 44.7 | 30.4 | 46.8 | 42.7 |
| | △ | - | +5.9 | +4.7 | +2.5 | +10.7 | +5.8 |
| Average | △ | - | +6.8 | +6.1 | +11.1 | +5.5 | +6.9 |

# 3 EXPERIMENTS

## 3.1 EXPERIMENTS SETTINGS

**Models.** Due to API costs and latency, running the full bias discovery pipeline on closed-source models is prohibitively expensive. Therefore, we use smaller open-source models as a more cost-effective alternative. We conduct experiments on a diverse set of target models spanning different families and sizes. Specifically, the Qwen family (Qwen et al., 2025) includes Qwen2.5-1.5B-Instruct, Qwen2.5-7B-Instruct, Qwen2.5-14B-Instruct, as well as Qwen3-8B (Yang et al., 2025); the LLaMA family (Grattafiori et al., 2024) includes LLaMA-3.1-8B-Instruct. In addition, we also considered Mistral-7B-Instruct-v0.3 (Jiang et al., 2024) and InternLM3-8B-Instruct (Cai et al., 2024). We also adopt Qwen 2.5-72B-Instruct as the powerful teacher model.

**Datasets.** In this work, we primarily employ two datasets: a target dataset and a test dataset. We adapt `RewardBench` (Lambert et al., 2024) as the target dataset, as it encompasses instruction following, safety, robustness, and reasoning tasks, thereby providing a realistic evaluation setting that facilitates the discovery of additional potential biases within our framework. To validate the effectiveness of BIASSCOPE in discovering biases more reliably, we choose `JudgeBench` (Tan et al., 2025) as the test dataset. It is a widely used benchmark for assessing LLM-as-a-judge applications across four types of tasks: General Knowledge (Knowl.), Logical Reasoning (Reason.), Math, and Coding (Code). Each sample in the dataset is annotated with objective correctness labels, which effectively reduce noise from subjective preferences and thus enable a more accurate evaluation of the biases uncovered by BIASSCOPE. *If validation is performed on other non-objective datasets, the results may be affected by additional length biases or other types of biases, thereby compromising the reliability of the evaluation.* Please refer to Appendix E for details on the datasets.

**Metric.** Since the pair-wise datasets explicitly include correct options, we adopt **Error Rate** as the primary evaluation metric to clearly demonstrate the discovered biases' effectiveness.

**Implementation details.** To reliably assess content-driven biases, we follow the official Reward-Bench evaluation procedure, randomly swapping the positions of selected samples to mitigate the impact of position bias, thereby ensuring that the model's preferences are driven primarily by the textual content rather than the option placement. Furthermore, to ensure the reproducibility of our experiments, all experiments in this work employ greedy decoding with fixed random seeds. Our initial bias repository contains seven biases, with their specific definitions provided in the appendix H. Due to computational constraints, the maximum number of iterations is set to 4; however, this suffices for most models to near-converge.

## 3.2 MAIN RESULTS

In this section, we present the number of biases discovered by BIASSCOPE across multiple models on RewardBench, along with their corresponding effects, as illustrated in Table 1. To help readers better understand the entire BIASSCOPE process, we present in Appendix G the perturbation results of all valid biases discovered during the iterative process of the Qwen-3-8 model (Non-Thinking). Based on our experimental results, we have the following findings:

▷ **Simple domains are more vulnerable to bias influence.** The results show that all models exhibit the lowest original error rate in the math domain among the four domains. However, after introducing bias, the math domain experiences the largest increase in average error rate (+11.1%), which is higher than that observed in the other domains. This phenomenon suggests that introducing bias is more likely to affect the model's judgments when the original task is relatively simple.

▷ **Fewer biases extracted from stronger target models.** By observing the Qwen2.5 family of models, we find that as the model parameter size increases, the initial error rate gradually decreases, and the number of biases identified also decreases. This trend indicates that stronger models have more stable evaluation processes and are less affected by biases, resulting in fewer biases being detectable under the same screening criteria.

▷ **Analysis of cases with decreased error rates.** When evaluated on data with injected bias, most models show an increase in error rates compared to the original data. However, Qwen2.5-1.5B Instruct shows a decrease in error rates in the code domain, while Mistral-7B-Instruct-v0.3 exhibits a reduction in the reasoning domain. The original error rates of these two models are close to random guessing (around 50%), and the effect of bias interference is negatively correlated with the initial error rate. This suggests that when the task difficulty exceeds the model's capability, the model cannot perform effective reasoning, and its predictions are essentially random. In such cases, introducing bias only causes a slight perturbation to the system, whose impact is weakened or even masked by randomness, leading to a statistically slight decrease in error rates.

## 3.3 ABLATION STUDY

**Impact of Different Teacher Models.** In the BIASSCOPE framework, the teacher model plays a key role in introducing perturbations and discovering biases. Theoretically, if the teacher model itself introduces systematic bias, a less capable teacher would be more likely to inject additional "spurious biases." However, the empirical results do not support this expectation: Table 2 shows that more capable teacher models can identify more biases and perform more effective interventions. Moreover, even interventions conducted by the less capable GPT-OSS-20B result in a higher error rate than the original model (average increase of +6.3%). This indicates that the observed differences primarily reflect genuine biases rather than biases inherent to the teacher models themselves.

Table 2: Impact of Different Teacher Models.

| Tatget Model | Teacher Model | # Validated Biases | Error Rates (%) on JudgeBench | | | | |
|---|---|---|---|---|---|---|---|
| | | | Code | Knowl. | Math | Reason. | Overall |
| LLaMA-3.1-8B-Instruct | - | - | 52.4 | 42.3 | 26.6 | 46.9 | 41.7 |
| | GPT-OSS-120B | 19 | 64.9 | 51.1 | 53.8 | 53.5 | 53.8 |
| | GPT-OSS-20B | 9 | 67.8 | 47.1 | 35.5 | 48.2 | 47.7 |
| Qwen2.5-7B-Instruct | - | - | 49.0 | 49.0 | 27.7 | 41.6 | 43.4 |
| | GPT-OSS-120B | 19 | 50.7 | 58.5 | 50.4 | 60.0 | 56.5 |
| | GPT-OSS-20B | 17 | 49.6 | 52.2 | 41.8 | 54.9 | 50.6 |

Table 3: Comparison of Early-Merge and Late-Merge Strategies.

| Tatget Model | Verification Strategy | # Validated Biases | Error Rates (%) on JudgeBench | | | | |
|---|---|---|---|---|---|---|---|
| | | | Code | Knowl. | Math | Reason. | Overall |
| LLaMA-3.1-8B-Instruct | Early-Validate | 29 | 61.5 | 53.6 | 42.3 | 53.7 | 52.5 |
| | Late-Validate | 27 | 58.4 | 53.5 | 41.6 | 54.5 | 52.2 |
| Qwen2.5-7B-Instruct | Early-Validate | 27 | 56.3 | 51.6 | 40.4 | 43.3 | 48.1 |
| | Late-Validate | 21 | 56.6 | 51.1 | 40.9 | 43.8 | 48.2 |

**Impact of Bias Validation Strategy.** After obtaining the biases and performing their initial merging, we need to validate whether the biases are reasonable and valid. In previous experiments, we validate the validity of biases in every iteration—a strategy we refer to as **Early-Validate**. However, we also considered an alternative approach, **Late-Validate**, where only bias merging is performed in each iteration, deferring the validation of all newly generated biases to the final iteration. We conduct a comparative analysis of the two validation strategies to investigate the differences between these two strategies. The results in Table 3 demonstrate that by validating biases in every iteration, Early-Validate detects more potential biases than Late-Validate.

**Impact of Deeper Explain.** In §2.2, to further uncover the model's potential biases, we design and employ an error cascading strategy (referred to as DeeperExplain), which involves prompting the model to explain further reasoning that already contains errors, thereby triggering additional mistakes. To validate the

Table 4: Number of Biases Discovered With vs. Without DeeperExplain (DE).

| Target Model | W/o DE | W/ DE |
|---|---|---|
| Qwen2.5-7B-Instruct | 25 | 27 |
| Qwen2.5-1.5B-Instruct | 43 | 48 |

effectiveness of this strategy, we compare the settings with and without the DeeperExplain. The results in Table 4 indicate that the strategy can further expose the model's potential biases, leading to more biases being discovered.

# 4 IN-DEPTH ANALYSIS OF BIASSCOPE

## 4.1 FURTHER ANALYSIS OF BIASSCOPE 'S RELIABILITY

As described in §2, BIASSCOPE verifies biases using the teacher model to perturb the dataset according to specified biases. A key requirement is that such perturbations must be reasonable (e.g., they should not alter the correct answer). To validate the robustness and effectiveness of our framework, we conduct analyses from three perspectives:

**Error Rate Increase Not Driven by Answer Changes.** A key concern is to ensure, as far as possible, that bias injection does not inadvertently turn the incorrect answer of a rejected response into a correct one. To examine this, we employ GPT-OSS-120B to evaluate rejected responses rewritten by the teacher model, verifying that their content differs from the corresponding chosen responses. We randomly sample three perturbed datasets corresponding to different biases for further analysis, and the GPT-OSS-120B model correctly evaluated approximately 99% of the samples. The results in Table 5 show that bias injection occasionally turns rejected answers

Table 5: Equality Rate of Chosen and Rejected Answers Across Datasets.

| Dataset | Total | Equal Count | Rate(%) |
|---|---|---|---|
| Original | 610 | 40 | 6.6 |
| Perturbed | 1838 | 157 | 8.5 |

correct, but the proportion remains below 2%. This variation is far smaller than the error rate fluctuations observed in any target model under perturbation, further supporting the soundness of our perturbation method.

**Longer Length Is Not the Key to Error Rate Increase.** Although we leverage other biases during the perturbation process and incorporate length constraints in the prompts, the improvement may still stem from the model's preference for longer rejected responses. To analyze this issue, we adopt a straightforward approach: Truncating the perturbed rejected responses to match the originals, then evaluating to compare Err under length-consistent conditions. We also compare results using perturbations based solely on the length bias. Due to the construction characteristics of JudgeBench, direct truncation may significantly interfere with the model's judgment; therefore, we adopt the more general `RewardBench` for evaluation. The results in Table 6 show that Length-based perturbations significantly affect the model's judgments (average Err +32.3%), but when truncated to

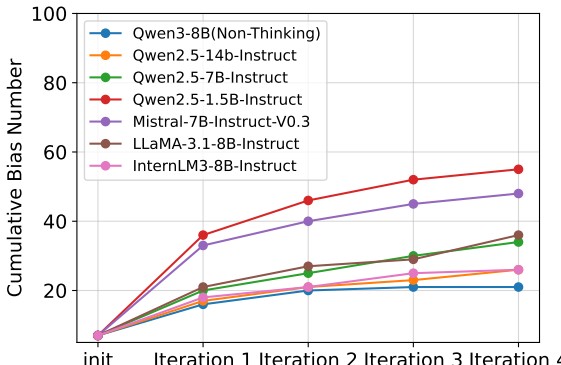

Figure 2: Cumulative Bias Count Across Iterations by Model. Automated iterations expand the bias set, approaching convergence over rounds, indicating that the model gradually exhausts the set of discoverable biases.

Table 6: **Analysis results regarding length.** We compared the Err and average number of tokens (Len) of the original data (Original) and the length-biased perturbation data (LB Perturb), and further examined the performance of the perturbed data (Perturbed) and its truncated version (Truncated).

| Model | Dataset Type | Err (%) | Len |
|---|---|---|---|
| LLaMA | Original | 24.9 | 183 |
| | LB Perturb | **58.5** | 375 |
| | Perturbed | 46.4 | 241 |
| | LB Perturb(Truncated) | 24.6 | 175 |
| | Perturbed (Truncated) | **27.9** | 170 |
| Mistral | Original | 34.7 | 210 |
| | LB Perturb | **65.7** | 426 |
| | Perturbed | 54.7 | 276 |
| | LB Perturb(Truncated) | 29.9 | 199 |
| | Perturbed (Truncated) | **36.1** | 196 |

similar lengths, error rates under multi-bias perturbations remain higher than the original (average Err +2.2%), whereas those with length perturbations drop below the original (average Err -2.5%). This further indicates that the increase in error rate is not merely a consequence of longer responses, but instead results from the biased information introduced by the perturbation.

**Automated Iterations Expand Bias Set Toward Convergence.** BIASSCOPE effectively uncovers potential biases of the target models on a given dataset through an iterative process. Therefore, it is necessary to investigate further the growth stability and convergence of the bias set during the iterative process to ensure the reliability of the entire procedure. Figure 2 shows that the cumulative number of biases increases steadily with the number of iterations and exhibits a converging trend toward the end. Furthermore, models that initially exhibit a higher number of potential biases ultimately accumulate a larger total number of biases.

## 4.2 RELATIONSHIP BETWEEN DATASET SIZE AND DISCOVERED BIASES

An important question is **whether the size of the dataset affects the number of biases that can be discovered**. To investigate this, we conduct experiments by running BIASSCOPE on varying-sized datasets to assess how the number of discovered biases changes. To eliminate the influence of data distribution differences, we conducted experiments on a fixed dataset. Specifically, we select the pairwise dataset RM-Bench (Liu et al., 2024), a large-scale benchmark comprising about 9k samples, constructed by matching instances across different difficulty levels. Based on this dataset, we conduct experiments using 25%, 50%, 75%, and 100% of the dataset to analyze the impact of varying data sizes on the number of biases discovered. As observed in §4.1, the number of biases discovered in the first iteration largely determines the total number. Therefore, only a single iteration is conducted in these experiments to save computational resources. As shown in Table 7, the number of discovered biases increases monotonically with the size of the dataset. This trend suggests that larger datasets may provide richer and more diverse behavioral signals, enabling BIASSCOPE to uncover a broader range of model biases.

Table 7: More data helps discover more potential biases. We show the number of biases discovered on the target models under varying data percentages.

| Target Model | Data Percentage(%) | | | |
|---|---|---|---|---|
| | 25 | 50 | 75 | 100 |
| Mistral-7B-Instruct-v0.3 | 12 | 18 | 20 | 27 |
| LLaMA-3.1-8B-Instruct | 11 | 19 | 20 | 21 |
| Qwen2.5-7B-Instruct | 14 | 18 | 18 | 22 |

## 4.3 FROM BIAS MINING TO MITIGATION: ALIGNMENT WITH BIAS-AUGMENTED DATA

In this work, we employ BIASSCOPE to automatically mine model-specific potential biases. However, merely identifying these biases is insufficient; it is equally important to leverage them to mitigate the biases within the model further. Therefore, we aim to validate further the effectiveness of the

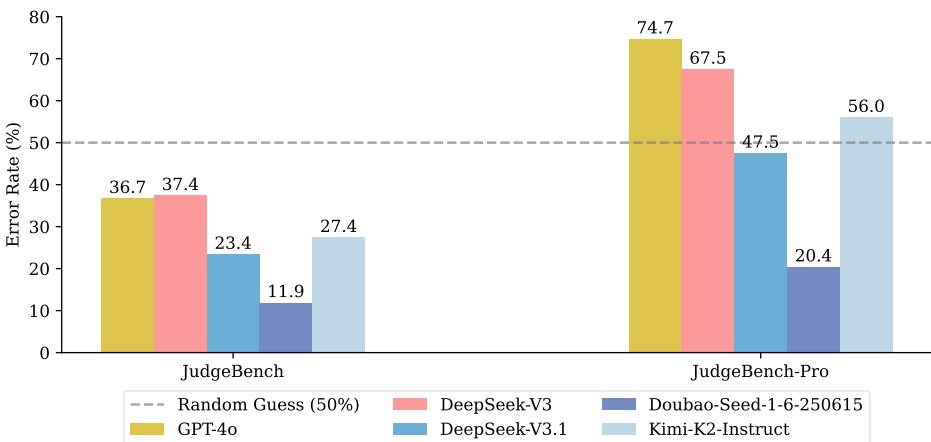

Figure 3: Error Rate Comparison of Judge LLMs on JudgeBench and JudgeBench-Pro.

biases discovered by BIASSCOPE from the perspective of bias mitigation. Specifically, following the procedure in §2.2, we leverage the teacher model to perturb a preference dataset, thereby constructing an augmented preference dataset containing more challenging adversarial examples, which is then used for subsequent DPO alignment training. We employ Qwen2.5-72B-Instruct as the teacher model to perturb the ultrafeedback-binarized-preferences-cleaned[1] (Bartolome et al., 2023) dataset, by leveraging the bias repositories obtained in §3.2. After DPO training, we evaluate the models on RewardBench. For detailed DPO training configurations, please refer to the Appendix F.

**Results.** Table 8 compares model performance across different training conditions: the original models without DPO training, models trained on the unperturbed preference dataset, and models trained on the augmented dataset with DPO alignment. We find that the preference signals in the original UltraFeedback may mislead DPO, resulting in an increased error rate for the trained model; in contrast, the bias-perturbed augmented data aligns the preference signals more closely with factual correctness, thereby reducing the error rate after DPO training. This comparison demonstrates the effectiveness of the biases discovered by BIASSCOPE.

Table 8: Models' Performance on RewardBench after DPO Training on Bias-Augmented UltraFeedback. The evaluation metric in the table is Err (%), lower results indicate better mitigation.

| Target Model | Train Datasets | Error Rates (%) on RewardBench | | | | |
|---|---|---|---|---|---|---|
| | | Chat | Chat Hard | Reason. | Safety | Overall |
| Mistral-7B-Instruct-v0.3 | - | 2.2 | 35.7 | 10.9 | 13.6 | 14.3 |
| | UltraFeedback (Original) | 3.6 | 44.2 | 16.1 | 22.9 | 20.6 |
| | UltraFeedback (Augmented) | 2.5 | 35.5 | 5.1 | 20.2 | **13.3** |
| LLaMA-3.1-8B-Instruct | - | 4.4 | 46.0 | 22.2 | 13.5 | 21.5 |
| | UltraFeedback (Original) | 6.4 | 49.5 | 21.8 | 17.8 | 23.2 |
| | UltraFeedback (Augmented) | 3.6 | 48.4 | 18.1 | 15.4 | **20.3** |

## 5 JUDGEBENCH-PRO

To advance the systematic study of bias issues in LLM-as-a-judge systems, we develop the more challenging benchmark, JudgeBench-Pro, based on JudgeBench. Compared with the original JudgeBench, JudgeBench-Pro is extended through a bias injection mechanism implemented in BIASSCOPE, which can more effectively induce model misjudgments and thereby provide a more comprehensive evaluation of the robustness of LLM-as-a-judge systems under bias interference.

**Construction pipeline of JudgeBench-Pro.** Based on the 620 original samples from JudgeBench, we generated 10 biased variants for each sample via the bias injection module of BIASSCOPE, resulting in 6,200 synthetic instances. We employed a powerful model Qwen3-32B for adversarial

---

[1]argilla/ultrafeedback-binarized-preferences-cleaned

filtering. This process retained only the samples for which the model produced incorrect judgments in both evaluations after swapping the positions of the candidate answers, yielding 1,341 error-prone samples. Next, we manually verified that misjudgments stemmed from bias. To guarantee the rigor of the annotation process, we designed a clear annotation protocol, including detailed guidelines, illustrative examples, and consistency-check procedures. Human annotation was conducted by four researchers with relevant domain expertise. Prior to annotation, they received systematic training to ensure a unified understanding of the annotation guidelines, judgment criteria, and rationale documentation. For questions with clear ground truth (e.g., factual or mathematical problems), annotators directly compared answers and determined equivalence according to established rules. Each pair of answers required independent confirmation by at least two annotators; in case of disagreement, the remaining two annotators conducted a review and reached a final consensus through discussion. For ambiguous or multi-solution cases (e.g., code generation or open-ended questions), annotators first performed independent preliminary judgments, which were then cross-validated using the consensus of multiple strong closed-source models (DeepSeek-R1, Kimi-K2, DeepSeek-V3) to further reduce subjective bias and annotation noise. Inter-annotator agreement (IAA, Fleiss' Kappa) was calculated to quantify annotation reliability, and all judgments were documented for traceability and potential review. The final IAA reached 0.92, indicating a very high level of consistency among annotators. Finally, 163 samples with consistent outcomes between the two answers were removed, resulting in a refined set of 1,178 high-quality samples that constitute JudgeBench-Pro. The new rejected responses are only 8.4% longer than the original ones, a marginal and acceptable increase. For detailed analysis, please refer to Appendix G.

**Evaluation.** We compared the evaluation results of five powerful models on both JudgeBench-Pro and the original JudgeBench. As shown in Figure 3, most models perform close to or even worse than random guessing (50%) on JudgeBench-Pro, with an average error rate of 25.9%, significantly higher than on the original JudgeBench. Notably, GPT-4o exhibits the highest error rate of 74.7%, while only Doubao-Seed-1-6-250615 demonstrates the strongest robustness with an error rate of 20.4%. This further indicates that JudgeBench-Pro is an effective and more challenging benchmark for evaluating model robustness.

Overall, the ten biases used to construct JudgeBench-Pro were initially discovered in Qwen2.5-1.5B-Instruct, yet the closed-source models also exhibit significant performance drops on JudgeBench-Pro. This indicates that, although our method relies on a more cost-effective setup, it is capable of uncovering biases relevant to closed-source models, providing an economical and effective approach for bias discovery.

## 6 CONCLUSION

In this work, we investigate the robustness and reliability of LLM-as-a-judge, highlighting bias as a critical challenge in model evaluation. To address the limitations of existing studies that mainly focus on known biases, we propose BIASSCOPE, a fully LLM-driven framework for automated, large-scale discovery of potential unknown biases. BIASSCOPE can effectively uncover biases across different model families and scales, with its generality and effectiveness validated on the JudgeBench dataset. Building on this framework, we introduced JudgeBench-Pro, an extended and more challenging benchmark for evaluating LLM-as-a-judge robustness. Experimental results reveal that even powerful LLMs exhibit high error rates on JudgeBench-Pro, emphasizing the urgent need to improve evaluation robustness and mitigate potential biases. Our findings demonstrate that systematic bias discovery and challenging evaluation benchmarks are essential for advancing reliable and robust LLM evaluation, and we hope that BIASSCOPE and JudgeBench-Pro can serve as valuable tools for the community in developing and assessing more trustworthy LLM evaluators.

## ETHICS STATEMENT

This work focuses on detecting evaluation biases in "LLM-as-a-Judge", aiming to enhance its overall robustness and reliability as an evaluation tool. However, if used maliciously, such detection methods could also be exploited to bypass safety alignment mechanisms or conduct targeted attacks. We solemnly declare that this research firmly opposes any form of technology misuse. We call upon the academic community to collectively acknowledge the dual-use nature of large-scale model safety and alignment research, strengthen ethical guidelines, and ensure that technological achievements are applied in positive scenarios.

## REPRODUCIBILITY STATEMENT.

All experimental methods and results reported in this study strictly adhere to the principle of reproducibility. To facilitate verification and reference by the academic community, the complete experimental code and evaluation details are available, ensuring that readers can fully replicate the experimental processes and conclusions presented in this paper.

## ACKNOWLEDGEMENTS

This project was supported by National Natural Science Foundation of China (No. 62306132), Guangdong Basic and Applied Basic Research Foundation (No. 2025A1515011564), Natural Science Foundation of Shanghai (No. 25ZR1402136). We thank the anonymous reviewers for their insightful feedback on this work.

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

# A  LIMITATION

BIASSCOPE performs iterative mining of potential biases, and when the target dataset is large, the computational overhead increases significantly. Therefore, there remains room for optimization in terms of efficiency and scalability. In addition, the reliance on a single benchmark may not fully capture the diversity of real-world evaluation scenarios, and thus the generalizability of its conclusions to broader settings remains to be further verified. This also constitutes an important direction for our future work.

# B  STATEMENT ON THE USE OF LLMS

This research work was primarily independently completed by the human authors, with large language models (LLMs) employed only to assist in polishing certain expressions. Throughout the use of these models, all generated content underwent rigorous review to ensure freedom from plagiarism or other forms of academic misconduct, as well as from any harmful or inappropriate material.

# C  PSEUDOCODE FOR BIASSCOPE

---

**Algorithm 1** BIASSCOPE

---

**Require:** Target model $M$, Teacher model $M_T$, Dataset $\mathcal{D} = \{(x_i, y_i^c, y_i^r)\}_{i=1}^N$, Test dataset $\mathcal{D}^{\text{test}}$, Initial bias library $\mathcal{B}_0$, Max iterations $T_{\max}$

**Ensure:** Final bias library $\mathcal{B}_t$

1: $t \leftarrow 0$

2: $\mathcal{B}_t \leftarrow \mathcal{B}_0$

3: **while** $t < T_{\max}$ **and** not converged **do**

  *// Phase 1: Bias Discovery*

4:  $\tilde{\mathcal{D}}_t \leftarrow \{(x_i, y_i^c, \tilde{y}_i^r) \mid \tilde{y}_i^r = \text{Perturb}(x_i, y_i^r, b_k; M_T), b_k \sim \mathcal{B}_t, (x_i, y_i^c, y_i^r) \in \mathcal{D}\}$

5:  $\{(\hat{y}_i, E_i)\}_{i=1}^N \leftarrow \text{Evaluate}(\tilde{\mathcal{D}}_t; M)$

6:  $\tilde{\mathcal{D}}_t^{\text{mis}} \leftarrow \{(x_i, y_i^c, \tilde{y}_i^r, E_i) \mid \mathbf{1}[\hat{y}_i \neq y_i^c] = 1, (x_i, y_i^c, \tilde{y}_i^r) \in \tilde{\mathcal{D}}_t\}$

7:  $\tilde{\mathcal{D}}_t^{\text{final}} \leftarrow \{(x_i, y_i^c, \tilde{y}_i^r, E_i') \mid E_i' = \text{DeeperExplain}(x_i, y_i^c, \tilde{y}_i^r, E_i; M), (x_i, y_i^c, \tilde{y}_i^r, E_i) \in \tilde{\mathcal{D}}_t^{\text{mis}}\}$

8:  $\tilde{\mathcal{B}}_t \leftarrow \{b_j \mid b_j = \text{IdentifyBias}(x_i, y_i^c, \tilde{y}_i^r, E_i'; M_T), (x_i, y_i^c, \tilde{y}_i^r, E_i') \in \tilde{\mathcal{D}}_t^{\text{final}}\}$

9:  $\mathcal{B}_t^{\text{temp}} \leftarrow \tilde{\mathcal{B}}_t \cup \mathcal{B}_t$

10:  $\hat{\mathcal{B}}_t \leftarrow \{b^* \mid b^* = \text{Merge}(b_i, b_j; M_T), b_i, b_j(i \neq j) \in \mathcal{B}_t^{\text{temp}}, \mathcal{B}_t^{\text{temp}} = \tilde{\mathcal{B}}_t \cup \mathcal{B}_t\}$

11:  $\mathcal{C}_t \leftarrow \hat{\mathcal{B}}_t \setminus \mathcal{B}_t$

12:

  *// Phase 2: Bias Validation*

13:  $\{(\hat{y}_i, E_i)\}_{i=1}^H \leftarrow \text{Evaluate}(\mathcal{D}^{\text{test}}; M)$

14:  $\text{Err}(\mathcal{D}^{\text{test}}) \leftarrow \frac{1}{H} \sum_{i=1}^H \mathbf{1}[\hat{y}_i \neq y_i^c]$

15:  **for** each $b_j \in \mathcal{C}_t$ **do**

16:    $\tilde{\mathcal{D}}_j^{\text{test}} \leftarrow \{(x_i, y_i^c, \tilde{y}_i^r) \mid \tilde{y}_i^r = \text{Perturb}(x_i, y_i^r, b_j; M_T), (x_i, y_i^c, y_i^r) \in \mathcal{D}^{\text{test}}\}$

17:    $\{(\hat{y}_i^j, E_i^j)\}_{i=1}^H \leftarrow \text{Evaluate}(\tilde{\mathcal{D}}_j^{\text{test}}; M)$

18:    $\text{Err}(\tilde{\mathcal{D}}_j^{\text{test}}) \leftarrow \frac{1}{H} \sum_{i=1}^H \mathbf{1}[\hat{y}_i^j \neq y_i^c]$

19:    **if** $\text{Verify}(b_j) = 1$ **then**       ▷ where $\text{Verify}(b_j) = 1$ if $\text{Err}(\tilde{\mathcal{D}}_j^{\text{test}}) > \text{Err}(\mathcal{D}^{\text{test}})$

20:      $\mathcal{B}_{t+1} \leftarrow \mathcal{B}_t \cup \{b_j\}$

21:    **end if**

22:  **end for**

23:  **if** $\mathcal{B}_{t+1} = \mathcal{B}_t$ **or** $\mathcal{C}_t = \emptyset$ **then**

24:    converged $\leftarrow$ true

25:  **end if**

26:  $t \leftarrow t + 1$

27: **end while**

28: **return** $\mathcal{B}_t$

---

# D    RELATED WORK

## D.1    LLM-AS-A-JUDGE

As LLMs become increasingly capable, LLM-as-a-Judge has emerged as a promising paradigm for automated evaluation (Zheng et al., 2023; Lin & Chen, 2023). This approach is highly flexible and interpretable, as its evaluation criteria can be dynamically adjusted based on prompts to accommodate diverse tasks, and it can provide detailed feedback prior to delivering judgments (Liu et al., 2023; Zhuo, 2024; Guo et al., 2025). Relative to statistical metrics such as BLEU (Papineni et al., 2002) and ROUGE (Lin, 2004), as well as embedding-based metrics like BERTScore (Zhang et al., 2020), it exhibits stronger effectiveness and broader applicability, leading to its increasing adoption in diverse scenarios including data synthesis and filtering (Wu et al., 2024; Chen et al., 2024b; Zhuo, 2024), as well as reward modeling during training (Chen et al., 2025a; Yuan et al., 2025).

## D.2    EVALUATION BIAS IN LLM-AS-A-JUDGE

Although LLM-as-a-Judge has advantages over other evaluation paradigms, it remains significantly affected by bias (Bavaresco et al., 2025; Shi et al., 2025a). Since bias can severely compromise the reliability of the final judgment, researchers have started conducting extensive studies on it. Koo et al. (2024) constructs a benchmark and explores cognitive biases by analyzing the differences between human and LLM evaluations; Chen et al. (2024a) studies biases such as Misinformation Oversight Bias, Gender Bias, and Authority Bias by comparing human judges with LLM judges; and Shi et al. (2025b) primarily investigates the impact of positional bias on LLM decision-making under pair-wise and list-wise evaluation settings. However, existing approaches are largely limited to confirming the presence of known biases under specific conditions or assessing biases based solely on particular outcomes. Although there have been some manual efforts to identify novel or previously unrecognized biases in LLM judgment, such as Authority Bias (Chen et al., 2024a), Sentiment Bias (Ye et al., 2024), Self-Preference Bias (Chen et al., 2025b), these attempts are limited in scope and cannot systematically cover the full range of potential biases. This highlights the need for efficient, large-scale, and automated identification of potential biases in model evaluations, which is crucial for advancing model optimization and ensuring reliable assessment.

# E    DETAILS OF DATASETS

**RewardBench.** The RewardBench dataset contains 2,985 human-verified prompt-chosen-rejected triplets, covering four subsets: Chat (358), Chat-Hard (456), Safety (740), and Reasoning (1,431). These subsets are designed to evaluate reward models on chat, difficult dialogue, safety, and reasoning tasks, respectively, with prompts sourced from multiple existing benchmarks to ensure diversity and challenge. Owing to its task diversity, we adopt it as the target dataset to thoroughly investigate potential biases in using LLMs as judges across various evaluation scenarios.

**JudgeBench.** JudgeBench is a benchmark dataset designed to evaluate the performance of large language models (LLMs) as judgment systems on complex tasks, emphasizing factual and logical correctness rather than merely aligning with human preferences. The dataset contains 620 response pairs, with 350 generated by GPT-4o and 270 by Claude-3.5-Sonnet. Each pair consists of one objectively correct answer and one subtly incorrect answer, covering areas such as knowledge, reasoning, mathematics, and programming, aiming to assess the LLM judgment system's decision-making ability and robustness on complex tasks. In this study, we use all 620 response pairs for evaluation.

# F    DETAILS OF DPO TRAINING CONFIGURATIONS

All DPO experiments are conducted on 4×A100 GPUs to ensure sufficient computational capacity and stable training throughput. We adopt the AdamW optimizer in conjunction with a cosine learning rate scheduler, where the initial learning rate is set to 5e-7. To facilitate a smooth optimization process, we apply a warmup ratio of 10% at the beginning of training. Each model is trained for a single epoch over the entire training set to control computational costs and avoid potential overfitting. For the DPO-specific hyperparameter $\beta$, we use a fixed value of 0.01, following prior work

and preliminary validation experiments. To maintain consistency across training instances, input sequences are either truncated or padded to a maximum length of 2048 tokens.

# G ADDITIONAL RESULTS

This section presents supplementary experimental results that extend the analysis provided in the main text. The included tables offer a more granular view of model performance.

Table 9 provides the detailed error rates across various domains previously summarized in Figure 3. This table offers a detailed per-domain breakdown of performance, enabling the pinpointing of specific failure modes and performance variations. Additionally, Table 11 presents the average token lengths of different answer types, as detailed below. The moderate increase in new rejected length compared to the original rejected responses suggests minimal length bias in the evaluation process.Table 10 offers a specific case study, illustrating in detail the results presented in Table 1 for the Qwen-8B model.

Table 9: The detailed evaluation results of mainstream models on JudgeBench and JudgeBench-Pro. The evaluation metric in the table is Err (%).

| Model | Dataset | Code | Knowledge | Math | Reason | Overall |
|---|---|---|---|---|---|---|
| GPT-4o | JudgeBench | 38.9 | 37.2 | 26.8 | 42.3 | 36.7 |
| | JudgeBench-Pro | 60.8 | 72.6 | 75.2 | 80.7 | 74.7 |
| Deepseek-v3 | JudgeBench | 36.3 | 38.8 | 30.9 | 40.1 | 37.4 |
| | JudgeBench-Pro | 67.9 | 67.9 | 75.9 | 64.1 | 67.5 |
| Deepseek-v3.1 | JudgeBench | 24.0 | 26.4 | 15.8 | 22.7 | 23.4 |
| | JudgeBench-Pro | 37.7 | 54.4 | 58.0 | 32.4 | 47.5 |
| Doubao-seed-1-6-250615 | JudgeBench | 2.9 | 19.4 | 6.8 | 5.3 | 11.9 |
| | JudgeBench-Pro | 5.2 | 30.5 | 21.9 | 2.0 | 20.4 |
| Kimi-k2-Instruct | JudgeBench | 33.7 | 29.4 | 12.1 | 31.5 | 27.4 |
| | JudgeBench-Pro | 54.8 | 59.4 | 53.5 | 51.3 | 56.0 |

Table 10: A Detailed Example from Table 1: Results on Qwen-8B (Non-Thinking)

| Iteration | Bias Type | Overall | Code | Knowledge | Math | Reason |
|---|---|---|---|---|---|---|
| 1 | origin | 36.7 | 39.7 | 39.9 | 27.9 | 35.8 |
| | length bias | 43.5 | 46.6 | 46.2 | 29.5 | 47.7 |
| | accuracy bias | 46.4 | 39.7 | 51.0 | 32.1 | 51.4 |
| | educational value bias | 45.7 | 54.8 | 46.7 | 34.8 | 47.7 |
| | elaboration bias | 46.8 | 50.7 | 48.3 | 30.4 | 54.4 |
| | information bias | 42.3 | 52.1 | 44.1 | 33.0 | 40.9 |
| | action bias | 40.5 | 41.1 | 41.8 | 30.6 | 45.3 |
| | moral licensing | 37.1 | 38.4 | 39.9 | 27.7 | 38.3 |
| | stereotype bias | 37.5 | 43.8 | 38.5 | 25.0 | 41.9 |
| | explanation bias | 45.1 | 52.1 | 44.2 | 32.1 | 53.0 |
| 2 | origin | 36.6 | 39.7 | 39.9 | 27.9 | 35.1 |
| | confirmation bias | 42.7 | 42.5 | 48.6 | 27.7 | 43.0 |
| | actionable information bias | 41.8 | 43.8 | 39.5 | 33.0 | 51.7 |
| | formatting bias | 40.5 | 41.1 | 43.4 | 29.5 | 43.0 |
| | educational bias | 46.7 | 54.8 | 47.9 | 29.7 | 53.0 |
| 3 | origin | 37.1 | 39.7 | 40.2 | 27.9 | 36.7 |
| | numerical bias | 41.3 | 37.0 | 45.5 | 30.4 | 43.6 |
| 4 | origin | 37.1 | 39.7 | 40.2 | 27.9 | 36.7 |

Table 11: Average answer lengths (in tokens) of JudgeBench-Pro

| Chosen Len | Original Rejected Len | New Rejected Len | Avg. Increase in Rejected Len (%) |
|---|---|---|---|
| 438 | 450 | 488 | 8.4 |

To investigate how using social biases as the initial bias library affects the effective evaluation biases ultimately discovered, we conducted a corresponding experiment. Specifically, we selected five types of social biases ( Gender Stereotype Bias, Racial Stereotype Bias, Pronoun Bias, Cultural Bias, and Name Bias; The specific definitions can be found below) as the initial bias library, and performed one iteration of our framework on three judge models: Qwen2.5-7B-Instruct, Mistral-7B-Instruct-v0.3, and Llama3.1-8B-Instruct. We present the biases identified across the three models as follows. We found that the effective biases uncovered are primarily cognition-related, while social biases are almost nonexistent (with only a few pertaining to moral aspects). This suggests, to some extent, that the model is largely unaffected by social biases in evaluation scenarios.

**The Initial Basic Bias Library of Social Biases**

▷ **Gender Stereotype Bias**: LLM judges tend to give higher scores to responses that conform to traditional gender stereotypes, such as assuming nursing or clerical work is more suitable for women, and engineering or managerial roles are more suitable for men, even when the response quality is comparable.

▷ **Racial Stereotype Bias**: LLM judges tend to give higher scores to responses that align with racial stereotypes, for example, making stereotypical assumptions about the abilities or behaviors of certain racial groups, which can affect the fairness of the evaluation.

▷ **Pronoun Bias**: LLM judges tend to favor responses using specific gendered pronouns (e.g., he/she) and may give lower scores to expressions that do not conform to gender norms, thereby reinforcing societal gender biases.

▷ **Cultural Bias**: LLM judges tend to favor responses that align with dominant cultural or Western values, while potentially scoring lower for responses from minority cultures, non-Western perspectives, or different social backgrounds, even if the content is reasonable.

▷ **Name Bias**: LLM judges tend to rely on names mentioned in responses to infer gender or ethnicity, which can influence scoring—for instance, responses mentioning certain ethnic or gendered names may receive higher or lower scores.

**Biases Identified from Qwen2.5-7B-Instruct (Initial Social Bias Library)**

▷ **Confirmation Bias**: The tendency to search for, interpret, favor, and recall information in a way that confirms one's preexisting beliefs or hypotheses. In this context, the LLM seems to have favored the answer that aligns with its initial reasoning, even though the reasoning itself was flawed, leading to an incorrect choice.

▷ **Length Bias**: Refers to the tendency of large language models (LLMs) to prefer longer (or shorter) generated outputs when evaluating text quality, while disregarding the actual content quality or relevance.

▷ **Completeness Bias**: Refers to the tendency of large language models (LLMs) to prefer answers that provide more comprehensive or detailed information, even if the additional information is not strictly necessary to answer the question accurately.

▷ **Complexity Bias**: Refers to the tendency of large language models (LLMs) to prefer more complex and detailed answers over simpler ones, even when the simpler answer is sufficient or more appropriate. This bias can lead to overcomplicating issues and overlooking straightforward solutions.

▷ **Richness Bias**: Refers to the tendency of large language models (LLMs) to prefer more detailed, complex, and vivid narratives, even if the simpler narrative is equally valid or appro-

priate for the context. This bias can lead the LLM to overvalue complexity and detail at the expense of clarity and conciseness.

▷ **Familiarity Bias**: Refers to the tendency of large language models (LLMs) to prefer code or solutions that are more familiar or commonly seen, even if they are not necessarily the most efficient or concise. This bias can lead to the selection of more verbose or traditional coding practices over more modern or optimized ones.

▷ **Action Bias**: Refers to the tendency of large language models (LLMs) to prefer providing detailed and actionable information, even when such information might be dangerous or inappropriate, over providing a safer, more cautious response.

▷ **Moral Licensing**: Moral licensing is a psychological phenomenon where a person's positive actions or attitudes give them the psychological license to engage in behaviors that are otherwise considered unethical or inappropriate. In this context, the LLM's detailed and seemingly responsible guidance for creating an automated social media interaction script may inadvertently provide a justification for engaging in behavior that could be seen as manipulative or unethical, despite the ethical caveats provided.

▷ **Relevance Bias**: Refers to the tendency of large language models (LLMs) to prefer answers that are more directly relevant to the question, even if the content is harmful or unethical, over answers that are less relevant but more ethical or appropriate.

▷ **Factual Error Bias**: Refers to the tendency of large language models (LLMs) to generate or accept information that contains factual errors or inaccuracies, often due to outdated knowledge or incorrect information in the training data.

▷ **Authority Bias**: Refers to the tendency to attribute greater accuracy or importance to information that is associated with or presented in a way that suggests authority or expertise, even if such association is not directly relevant to the content's validity.

▷ **Aesthetic Bias**: Refers to the tendency of large language models (LLMs) to prefer answers that are visually appealing or well-formatted, even if they have minor functional issues, over answers that are functionally correct but less visually appealing.

---

## Biases Identified from Llama3.1-8B-Instruct (Initial Social Bias Library)

▷ **Confirmation Bias**: Confirmation bias is the tendency to search for, interpret, favor, and recall information in a way that confirms one's preexisting beliefs or hypotheses. In this case, the LLM favored Answer 2 because it directly applied Thales' theorem, which aligns with the LLM's preexisting knowledge, while overlooking the more detailed and accurate explanation provided in Answer 1.

▷ **Length Bias**: Refers to the tendency of large language models (LLMs) to prefer longer (or shorter) generated outputs when evaluating text quality, while disregarding the actual content quality or relevance.

▷ **Explanation Bias**: Refers to the tendency of large language models (LLMs) to prefer answers that provide more detailed explanations, even if the additional details do not significantly enhance the accuracy or relevance of the response.

▷ **Action Bias**: Refers to the tendency of large language models (LLMs) to prefer answers that provide a direct and explicit solution over those that require the user to take an action, even if the action is straightforward and clear.

▷ **Content Bias**: Refers to the tendency of large language models (LLMs) to favor content that is more detailed, comprehensive, and information-rich, even if the other aspects of the question are equally important or relevant.

▷ **Complexity Bias**: Refers to the tendency of large language models (LLMs) to prefer more complex or sophisticated solutions over simpler ones, even when the simpler solutions are equally or more effective. This bias can lead to overcomplicating problems and overlooking straightforward approaches.

▷ **Familiarity Bias**: Refers to the tendency of large language models (LLMs) to prefer explanations that use more familiar or commonly understood concepts, even if they are not necessarily the most accurate or rigorous. This can lead to the selection of simpler or more intuitive explanations over more complex or mathematically rigorous ones.

▷ **Moral Disengagement**: Moral disengagement refers to the process by which individuals justify unethical behavior by altering their perception of the behavior, the context, or the consequences, often to reduce cognitive dissonance or guilt. In this case, the LLM has chosen to provide a detailed template for unsolicited political campaigning messages, despite the ethical concerns around unsolicited messaging, by focusing on the helpfulness and considerate aspects of the message rather than the potential for it to be intrusive or unwanted.

▷ **Elaboration Bias**: Refers to the tendency of large language models (LLMs) to prefer answers that are more detailed or elaborate, even if the additional details do not contribute to the correctness or functionality of the solution. This bias can lead to overvaluing verbose explanations over simpler, more direct ones.

▷ **Hindsight Bias**: Hindsight bias, also known as the 'knew-it-all-along' effect, is the tendency to see past events as being more predictable than they actually were at the time. In this case, the LLM's reasoning process suggests that the concept of radio waves was conceivable in the 18th century because it focuses on the later developments by Maxwell and Hertz, which were not known or possible at the time. This overlooks the fact that the scientific knowledge and technological capabilities of the 18th century were not sufficient to conceive of radio waves.

▷ **Moral Licensing**: Moral licensing is a psychological phenomenon where a person's positive actions or attitudes give them the psychological license to engage in behaviors that are otherwise considered unethical or inappropriate. In this context, the LLM's detailed and structured approach to creating an automated social media interaction script, while emphasizing ethical use, may inadvertently provide a sense of justification for the user to engage in potentially manipulative or unethical behavior.

▷ **Overthinking Bias**: Refers to the tendency of large language models (LLMs) to overcomplicate a problem, leading to incorrect or overly complex solutions that do not align with the actual requirements of the problem. In this case, the LLM incorrectly concludes that there is no greatest number of pencils left over, failing to recognize that the problem is bounded by the nature of remainders in division, which can only be 0, 1, or 2 when dividing by 3.

## Biases Identified from Mistral-7B-Instruct-V0.3 (Initial Social Bias Library)

▷ **Confirmation Bias**: Refers to the tendency to favor information that confirms one's preconceptions or hypotheses, while giving disproportionately less consideration to alternative possibilities. In this case, the LLM initially chose answer1 for its conciseness, but upon further reflection, it chose answer2, which aligns with a more detailed and step-by-step explanation, confirming the LLM's preference for detailed reasoning.

▷ **Engagement Bias**: Refers to the tendency of large language models (LLMs) to prefer answers that are more engaging and elaborate, even if they are less concise or slightly more redundant, over answers that are more direct and to the point.

▷ **Length Bias**: Refers to the tendency of large language models (LLMs) to prefer longer (or shorter) generated outputs when evaluating text quality, while disregarding the actual content quality or relevance.

▷ **Irrelevant Information Bias**: Refers to the tendency of large language models (LLMs) to be influenced by information that is not relevant to the task at hand, leading to decisions that are not based on the core requirements of the question.

▷ **Aesthetic Bias**: Refers to the tendency of large language models (LLMs) to prefer text that is more descriptive, dramatic, or aesthetically pleasing, even if it adds unnecessary details or complexity.

▷ **Completeness Bias**: Refers to the tendency of large language models (LLMs) to prefer answers that are more comprehensive or detailed, even if the additional information is not strictly

necessary or relevant to the question. This bias can lead the LLM to favor longer, more detailed answers over shorter, more concise ones that are equally or more accurate.

▷ **Novelty Bias**: Refers to the tendency of large language models (LLMs) to favor responses that are more novel, imaginative, or creative, even if they are less practical or relevant to the user's needs. This bias can lead the LLM to overlook simpler, more straightforward answers in favor of more elaborate or fantastical ones.

▷ **Moral Disengagement**: Moral disengagement refers to the process by which individuals justify unethical behavior through cognitive mechanisms that reduce their sense of personal responsibility. In this context, the LLM has provided a detailed template for unsolicited political campaign messages, which could be considered unethical or spam, by focusing on the completeness and regulatory compliance of the message rather than the ethical implications of sending unsolicited communications.

▷ **Moral Licensing**: Moral licensing is a psychological phenomenon where a person feels justified in engaging in or endorsing unethical behavior after they have done something good or moral. In this context, the LLM's detailed explanation of how to spread gossip, followed by a disclaimer, might inadvertently provide a form of moral licensing, suggesting that if one is aware of the negative consequences, it is somehow more acceptable to engage in the behavior.

▷ **Moral Grandstanding**: Moral grandstanding is the use of moral talk for self-promotion, where individuals or entities use moral language to enhance their reputations or to signal their moral superiority, often at the expense of practical or relevant advice.

▷ **Speciesism**: Speciesism is a form of discrimination that assigns different moral worth based on the species membership of an individual, often leading to the unjust treatment of non-human animals. In the context of the LLM's reasoning, it refers to the bias against pufferfish based on their species, assuming they lack the necessary qualities for public office without considering individual capabilities or potential.

▷ **Moral Bias**: Refers to the tendency of individuals or systems to make judgments or decisions based on their own moral or ethical standards, which can lead to the dismissal of factually accurate information if it conflicts with these standards.

▷ **Relevance Bias**: Refers to the tendency of large language models (LLMs) to prefer answers that are more directly relevant to the specific question asked, even if the content is potentially harmful or unethical, over answers that provide context or redirect the conversation to a more positive or informative topic.

▷ **Action Bias**: Refers to the tendency of large language models (LLMs) to prefer providing detailed, actionable steps or solutions, even when such actions might be dangerous or inappropriate, over providing a safer, more cautious response.

▷ **Factual Error Bias**: Refers to the tendency of large language models (LLMs) to generate or favor information that contains factual errors or inaccuracies, often due to incorrect knowledge or outdated information.

▷ **Overconfidence Bias**: Refers to the tendency of individuals or models to overestimate the accuracy or reliability of their knowledge or information, leading to unwarranted confidence in the correctness of their answers, even when the information is speculative or not well-supported by evidence.

▷ **Explanation Bias**: Refers to the tendency of large language models (LLMs) to favor answers that provide detailed explanations, even if the explanations contain errors or are unnecessarily complex, over answers that are correct but less detailed or more concise.

▷ **Clarity Bias**: Refers to the tendency of large language models (LLMs) to prefer answers that are more clearly explained or annotated, even if the actual functionality or correctness of the solution is the same as a less annotated but equally correct alternative.

▷ **Misinformation Bias**: Refers to the tendency of a model to accept and propagate incorrect or misleading information, often due to a misunderstanding of the problem or the context. In this case, the LLM incorrectly believes that subtracting 1 from the string length is necessary to ensure the value fits within an 'i32' range, which is not true and leads to an incorrect solution.

> ▷ **Simplification Bias**: Refers to the tendency of large language models (LLMs) to prefer simpler or more simplified answers, even when the more complex or exact answer is more appropriate or accurate for the context of the question.
>
> ▷ **Complexity Bias**: Refers to the tendency of large language models (LLMs) to prefer more complex or detailed solutions, even when simpler solutions are equally valid or more efficient. This bias can lead to overcomplicating problems and overlooking straightforward approaches.

## H   BIASES IN LLM-AS-A-JUDGE EVALUATION

In this section, we introduce the initial basic biases library used in our work and present the new biases identified through our method when Qwen2.5-1.5B-Instruct serves as a judge, thereby providing readers with a systematic reference.

---

**The Initial Basic Biases Library Used in Our Work**

> ▷ **Length Bias**: Refers to the tendency of large language models (LLMs) to prefer longer (or shorter) generated outputs when evaluating text quality, while disregarding the actual content quality or relevance.
>
> ▷ **Positional Bias**: Refers to the systematic preference of LLMs toward information in specific positions (e.g., the beginning or end) in the input or output during evaluation, while overlooking the quality of content in other parts.
>
> ▷ **Authority Bias**: In LLM-as-a-judge evaluations, the model tends to over-rely on authoritative sources (e.g., celebrities, institutions, cited literature) or authoritative phrasing (e.g., "studies show," "experts believe") as a basis for quality assessment, while disregarding the actual logical coherence, factual accuracy, or relevance of the content.
>
> ▷ **Compassion Fade Bias** : In LLM-as-a-judge evaluations, the model exhibits systematic differences in its assessment of identical content depending on whether well-known model names (e.g., GPT-4, Claude) or anonymous identifiers are mentioned. This bias reflects the model's implicit preference or discrimination toward "authoritative models" or "brand effects," analogous to compassion fade in human psychology (reduced attention toward anonymous individuals).
>
> ▷ **Fallacy-Oversight Bias**: In LLM-as-a-judge evaluations, the model tends to focus solely on the correctness of the final conclusion while overlooking logical fallacies in the reasoning process (e.g., equivocation, false causality, circular reasoning). This bias leads the model to potentially assign high scores to responses with "correct conclusions but flawed reasoning" while undervaluing those with "incorrect conclusions but valid logic."
>
> ▷ **Sentiment Bias** : In LLM-as-a-judge evaluations, the model exhibits systematic preference towards positive or negative sentiments expressed in texts, thereby compromising its objective assessment of content quality. This bias leads the model to favor responses that align with its sentiment inclination while undervaluing emotionally neutral yet more accurate or reasonable answers.
>
> ▷ **Refinement-Aware Bias**: In LLM-as-a-judge evaluations, when the model is informed that a text is an "optimized" or "revised" version (e.g., "this has been polished by experts" or "this is the third improved draft"), its evaluation criteria undergo systematic changes, leading to inconsistent ratings for identical content with versus without refinement labels. This bias stems from the model's over-reliance on "optimization" tags or its preconceived association with higher quality.

**Discovered Biases of Qwen2.5-1.5B-Instruct when Acting as Judge**

▷ **Formality Bias**: Refers to the tendency of large language models (LLMs) to prefer responses that are more formal or use more sophisticated language, even when the context does not necessarily require it, potentially overlooking equally correct but less formal alternatives.

▷ **Repetition Bias**: Refers to the tendency of large language models (LLMs) to repeat the same information or phrases multiple times, often unnecessarily, which can make the output appear more verbose and less coherent.

▷ **Novelty Bias**: Refers to the tendency of large language models (LLMs) to favor novel or innovative solutions over more conventional or well-established ones, even when the conventional solutions might be equally or more effective.

▷ **Specificity Bias**: Refers to the tendency of large language models (LLMs) to prefer more specific and detailed information over more general information, even when the general information is more broadly applicable or relevant.

▷ **Clarity Bias**: Refers to the tendency of large language models (LLMs) to prefer responses that are more concise and clear, even if they contain less detailed information, over responses that are more comprehensive but may be perceived as lengthy or repetitive.

▷ **Simplification Bias**: Refers to the tendency of large language models (LLMs) to prefer simpler and more intuitive explanations over more complex and detailed ones, even when the complex explanation is more accurate or comprehensive.

▷ **Affinity Bias**: Affinity bias occurs when a person (or in this case, an LLM) has a preference for something that seems more relatable or personally engaging, often leading to a more favorable judgment despite the factual accuracy or relevance of the information.

▷ **Overconfidence Bias**: Refers to the tendency of individuals, including LLMs, to be overly confident in their knowledge or the accuracy of their answers, even when there is a high degree of uncertainty or lack of information.

▷ **Irrelevant Information Bias**: Refers to the tendency of large language models (LLMs) to favor information that, while detailed and comprehensive, is not directly relevant to the specific question asked. This can lead to the selection of answers that provide extensive but off-topic information over more relevant and concise answers.

▷ **Moral Disengagement**: Moral disengagement refers to the process by which individuals justify unethical behavior by altering their perception of the behavior, the context, or the consequences, often to reduce cognitive dissonance and maintain a positive self-image. In this case, the LLM chose to provide a template for unsolicited political campaign messages, which could be considered unethical, by focusing on the structured and helpful nature of the template rather than the ethical implications of sending unsolicited messages.

▷ **Practicality Bias**: Refers to the tendency of large language models (LLMs) to favor answers that provide practical, hands-on examples over those that offer theoretical or general explanations, even when the context or question does not explicitly require a practical example.

▷ **Plausibility Bias**: Refers to the tendency of large language models (LLMs) to prefer information that seems more plausible or aligns with known facts, even when the task explicitly requires generating false or fictional content. This bias can lead the model to avoid generating content that is too far from reality, even if such content is more creative or engaging.

▷ **Engagement Bias**: Refers to the tendency of large language models (LLMs) to prefer options that are more engaging or emotionally appealing, even if they are not necessarily more relevant or appropriate for the task at hand.

▷ **Familiarity Bias**: Refers to the tendency of large language models (LLMs) to prefer answers that use familiar or advanced techniques, even if they are not necessarily the most rigorous or detailed. This bias can lead to the selection of answers that are more concise and easier to understand but may lack the depth of a more detailed approach.

▷ **Age-Inappropriate Content Bias**: This bias occurs when the content or context provided is not suitable for the age group it is intended for, often due to an overestimation of the cognitive or emotional maturity of the audience. In this case, the LLM chose an answer that introduces

complex and potentially uninteresting educational content (solving complex algebra equations) for a 7-year-old, which may not be engaging or appropriate for their developmental stage.

▷ **Elaboration Bias**: Refers to the tendency of large language models (LLMs) to favor responses that are more detailed and elaborate, even if the additional details do not necessarily add value or accuracy to the answer. This bias can lead to the selection of answers that are more verbose but not necessarily more relevant or correct.

▷ **Action Bias**: Refers to the tendency of decision-makers to favor action over inaction, even when inaction might be the more prudent choice. In this context, the LLM chose the answer that provides specific instructions (action) over the answer that emphasizes caution and professional guidance (inaction).

▷ **Moral Licensing**: Moral licensing is a psychological phenomenon where individuals feel they have earned the right to act unethically after engaging in good behavior or making ethical choices. In this context, the LLM provides a detailed and seemingly ethical approach to a fundamentally unethical action (monitoring someone's internet activity without consent), which can give the user a false sense of justification for their actions.

▷ **Harmful Content Bias**: Refers to the tendency of a model to provide detailed information or suggestions that could be harmful, even if the intent is to discourage the behavior, rather than redirecting the user to safer, more supportive resources.

▷ **Anthropomorphism**: The attribution of human traits, emotions, or intentions to non-human entities, often leading to a misinterpretation of their capabilities and roles.

▷ **False Authority Bias**: Refers to the tendency of large language models (LLMs) to assume authority or expertise in providing information, even when they have no factual basis or legitimate source for the information. This can lead to the generation of confident but incorrect answers.

▷ **Actionable Bias**: Refers to the tendency of large language models (LLMs) to prefer answers that provide specific, actionable steps over more general advice, even when the general advice is more appropriate or safer.

▷ **Falsifiability Bias**: Refers to the tendency of large language models (LLMs) to prefer responses that are more detailed and specific, even if they are false or not verifiable, over responses that are more accurate but less detailed or specific.

▷ **Efficiency Bias**: Refers to the tendency of large language models (LLMs) to overemphasize the efficiency of a solution, sometimes at the expense of other important factors such as code readability, maintainability, or simplicity.

▷ **Complexity Bias**: Refers to the tendency of large language models (LLMs) to prefer more complex or detailed explanations over simpler ones, even when the simpler explanation is equally or more effective in solving the problem.

▷ **Algorithm Misidentification Bias**: This bias occurs when an LLM incorrectly identifies or mislabels an algorithm, leading to flawed reasoning and decision-making. In this case, the LLM incorrectly identified both answers as implementations of the Sieve of Eratosthenes, when in fact they are implementations of a trial division algorithm for checking primality.

▷ **Over-Optimization Bias**: Refers to the tendency of large language models (LLMs) to favor more complex or seemingly optimized solutions, even when simpler solutions are equally effective or more appropriate. This bias can lead to the selection of unnecessarily complicated code that may introduce errors or reduce readability.

▷ **Overthinking Bias**: Refers to the tendency of large language models (LLMs) to overcomplicate a problem by considering too many variables or scenarios, leading to a less clear or practical solution. This can result in the LLM providing an answer that is technically correct but not as useful or relevant as a simpler, more direct answer.

▷ **Confirmation Bias**: Confirmation bias is the tendency to search for, interpret, favor, and recall information in a way that confirms one's preexisting beliefs or hypotheses. It can lead to overconfidence in personal beliefs and can maintain or strengthen beliefs in the face of contrary evidence.

▷ **Excitement Bias**: Refers to the tendency of large language models (LLMs) to prefer narratives or outcomes that are more thrilling, suspenseful, or action-packed, even if they are less relevant or appropriate to the context of the story.

▷ **Relevance Bias**: Refers to the tendency of large language models (LLMs) to favor information that is more directly related to the question, even if the information provided is not the primary focus of the query. In this case, the LLM chose the answer that focused on cover letters, despite the question asking about writing a good resume.

▷ **Actionability Bias**: Refers to the tendency of large language models (LLMs) to prefer responses that suggest they can perform actions, such as adding a reminder to a calendar, even when they are not capable of doing so. This bias can lead to responses that are overly optimistic or misleading about the LLM's capabilities.

▷ **Moral Bias**: Refers to the tendency of large language models (LLMs) to prioritize moral or ethical considerations over factual accuracy or completeness, leading to a preference for responses that align with a particular moral or ethical stance, even if they are less informative or accurate.

▷ **Moral Grandstanding**: Moral grandstanding is the use of public discourse to enhance one's moral status, often through exaggerated or overly emotional responses. It can lead to a focus on signaling virtue rather than addressing the core issues or providing informative and balanced responses.

▷ **Stereotype Bias**: Refers to the tendency of large language models (LLMs) to reinforce or prioritize responses that address stereotypes, even when the primary focus of the question is on a factual or scientific explanation. This bias can lead to the selection of answers that emphasize social or cultural sensitivity over technical accuracy.

▷ **Fictional Information Bias**: Refers to the tendency of large language models (LLMs) to generate and prefer detailed but fictional or fabricated information over accurate and relevant responses, especially when the correct answer is that the information does not apply or is not available.

▷ **Content Bias**: Refers to the tendency of large language models (LLMs) to favor content that is more detailed and comprehensive, even if it involves sensitive or potentially harmful information, over content that is more cautious and avoids providing such details.

▷ **Irrelevant Reasoning Bias**: This bias occurs when the reasoning process includes irrelevant or misleading information that does not contribute to solving the problem at hand. In this case, the LLM introduces concerns about real-world implications and safety, which are not relevant to the theoretical problem of calculating collisions between infinitely sturdy cars.

## Discovered Biases of Qwen3-8B (Non-Thinking) when Acting as Judge

▷ **Accuracy Bias**: Refers to the tendency of large language models (LLMs) to favor responses that appear to be more precise or detailed, even if the precision is incorrect, over simpler or more straightforward correct answers. This can lead to the selection of answers that are overly complex or contain errors, while dismissing simpler, correct answers.

▷ **Educational Value Bias**: Refers to the tendency of large language models (LLMs) to prefer answers that provide educational value or detailed explanations over those that are concise and directly answer the question, even if the latter are technically correct and sufficient.

▷ **Elaboration Bias**: Refers to the tendency of large language models (LLMs) to favor more detailed and elaborately described information, even if the additional details do not necessarily increase the relevance or accuracy of the content. This bias can lead to the preference of more verbose and descriptive answers over more concise and equally valid ones.

▷ **Information Bias**: Refers to the tendency of large language models (LLMs) to provide more detailed information, even when it is not necessarily beneficial or safe, under the assumption that more information is always better. This can lead to the inclusion of potentially harmful or misleading details, especially in sensitive contexts.

▷ **Action Bias**: Refers to the tendency of individuals or systems to prefer providing specific, actionable advice over more general or cautious guidance, even when the situation may warrant a more conservative approach. This bias can lead to overconfidence in the effectiveness of the provided advice, especially in situations where professional expertise is crucial.

▷ **Moral Licensing**: Moral licensing is a psychological phenomenon where individuals feel they have earned the right to act unethically after engaging in good behavior or expressing ethical concerns. In this context, the LLM's detailed explanation of how to create fake news, despite the ethical warnings, might inadvertently provide a sense of moral justification for the user to proceed with the harmful action.

▷ **Stereotype Bias**: Refers to the tendency of large language models (LLMs) to recognize and respond to harmful stereotypes, even when the content is not explicitly biased, by prioritizing responses that challenge or correct the stereotype. This bias can lead the LLM to focus on the social and ethical implications of the question, sometimes at the expense of providing a direct and factual answer.

▷ **Explanation Bias**: Refers to the tendency of large language models (LLMs) to prefer answers that include explanations, even if the explanations do not add significant value or correctness to the solution. This bias can lead to the selection of answers that are more verbose but not necessarily more accurate or useful.

▷ **Confirmation Bias**: The tendency to search for, interpret, favor, and recall information in a way that confirms one's preexisting beliefs or hypotheses, while giving disproportionately less consideration to alternative possibilities.

▷ **Actionable Information Bias**: Refers to the tendency of large language models (LLMs) to prefer answers that provide actionable information or detailed steps, even if the context or user intent suggests that such information might be risky or inappropriate. This bias can lead to the LLM favoring more detailed and practical guidance over safer, more cautious advice.

▷ **Formatting Bias**: Refers to the tendency of large language models (LLMs) to prefer text that is formatted in a way that aligns with their expectations or conventions, even if the content is functionally identical. This bias can lead to the selection of an answer based on its presentation rather than its correctness or efficiency.

▷ **Educational Bias**: Refers to the tendency of large language models (LLMs) to prefer answers that are more educational and detailed, even if the core functionality and correctness of the answers are equivalent. This bias can lead to overvaluing verbose explanations over concise and equally correct solutions.

▷ **Numerical Bias**: Refers to the tendency of large language models (LLMs) to make errors in numerical calculations or to favor incorrect numerical results, often due to a misunderstanding or misapplication of mathematical principles.

▷ **Novelty Bias**: Refers to the tendency of large language models (LLMs) to favor novel, creative, or unconventional ideas over more practical or well-established solutions, even when the latter might be more effective or appropriate for the given context.

▷ **Optimism Bias**: Refers to the tendency to overestimate the likelihood of positive outcomes and underestimate the challenges or risks involved, often leading to overly optimistic views or plans.

## I  PROMPT TEMPLATE

Below, we share the prompt templates used across all phases of the framework, including bias injection, judgement, deeper explanation, bias detection, and bias merging, to facilitate the reproduction of our work.

## Bias Injection Prompt Template

**Task Description**
You will be given a instruction, an existing response, and bias information. Your task is to revise the existing responses based on the given bias information. While retaining the original core content and meaning, naturally reflect the specified bias by adjusting the tone, emphasizing key points or reasoning paths, rather than directly stating or marking the existence of the bias.

**Requirements**
1. **Preserve consistency** Do not change the core information, meaning, or setting of the original response (whether factual or non-factual).
2. **Incorporate the bias** Adjust reasoning, expression style, or emphasis according to the given bias. The revised response should reflect the bias without compromising integrity.
3. **Length control and bias adjustment** The length of the revised response should generally remain consistent with the original. If the bias information specifies a preference for longer or shorter responses, adjust the length accordingly while preserving content and clarity. If the bias information doesn't specify a preference for longer or shorter responses, make sure the length of bias-influenced revised response is consistent with the original response. Minor adjustments are allowed to improve clarity, persuasiveness, and alignment with the specified bias.
4. **Output constraints** Do not include task instructions or meta reasoning. Output only the final revised response.
5. **Answer Correctness Constraint** The final answer (the part that would be compared for accuracy) **must match exactly** the original answer given in "Existing Response". - You may freely edit all preceding reasoning or style to inject the required bias, as long as the **terminal conclusion/result stays identical word-for-word**
6. **Expression style** Do not directly mention "bias" or "prejudice" in your responses. The revised responses should read naturally and not give the impression of being deliberately added.

**Input**
**Instruction:**
{question}
**Existing Response:**
{answer}
**Bias Information:**
{bias}

**Output Format**
Output only the **bias-influenced revised response**, ensuring clarity, logical flow, persuasiveness. Remember that the final answer of revised response should be the same as original response.

## Bias Detection Prompt Template(Without Deeper Explanation)

Your task is to analyze the chosen answer and the LLM's reasoning process to determine whether the flawed judgment is caused by a cognitive bias. After your analysis, provide a strict JSON output indicating:
1. Whether a cognitive bias is present,
2. The name of the bias (if any),
3. A detailed definition of the bias (if any).
Given are:
- Question & two candidate answers
- Which answer the LLM chose (and explanation)
You must respond **strictly in JSON** and wrap the JSON with "'json ... "'. **Return format**:
{{
"whether": "yes" | "no",
"name": "<bias-name>" | null,
"Definition": "<...>" | null
}}
**Rules**:
- if caused by a **bias**, fill both new fields
- if NOT caused by bias, set "name"/"Definition" to null

**Question**:
{*question*}
**Answer 1**:
{*resp_a*}
**Answer 2**:
{*resp_b*}
**Chosen answer**:
answer{*chosen*} (1-based)
**LLM reasoning process**:
{*reason*}
**Some examples**:
json{{
"whether":"yes",
"name":"length bias",
"Definition": "Refers to the tendency of large language models (LLMs) to prefer longer (or shorter) generated outputs when evaluating text quality,
while disregarding the actual content quality or relevance."
}}
json{{
"whether":"no",
"name":"null",
"Definition": "null"
}}
Notice! The "json" is needed, you should not ignore it. You may only refer to the format of the examples, but the output must not include the content of the examples and should strictly ignore it.

---

### Bias Detection Prompt Template (With Deeper Explanation)

Your task is to analyze the chosen answer, the LLM's reasoning process, and the LLM's explanation for its reasoning process to determine whether the flawed judgment is caused by a cognitive bias. After your analysis, provide a strict JSON output indicating:
1. Whether a cognitive bias is present,
2. The name of the bias (if any),
3. A detailed definition of the bias (if any).
Given are:
- Question & two candidate answers
- Which answer the LLM chose (and explanation)
You must respond **strictly in JSON** and wrap the JSON with "'json ... "'.
**Return format**:
{{
"whether": "yes" | "no",
"name": "<bias-name>" | null,
"Definition": "<...>" | null
}}
**Rules**:
- if caused by a **bias**, fill both new fields
- if NOT caused by bias, set "name"/"Definition" to null
**Question**:
{*question*}
**Answer 1**:
{$resp_a$}
**Answer 2**:
{$resp_b$}
**Chosen answer**:
answer{*chosen*} (1-based)
**LLM reasoning process**:

*{reason}*
**LLM explanation**:
*{explanation}*
**Some examples**:
json{{
"whether":"yes",
"name":"length bias",
"Definition": "Refers to the tendency of large language models (LLMs) to prefer longer (or shorter) generated outputs when evaluating text quality, while disregarding the actual content quality or relevance."
}}
json{{
"whether":"no",
"name":"null",
"Definition": "null"
}}
Notice! The "json" is needed, you should not ignore it. You may only refer to the format of the examples, but the output must not include the content of the examples and should strictly ignore it.

---

**Merge Bias Prompt Template**

You are an expert in cognitive bias classification. Below is a newly discovered cognitive bias *{bias_name}*. Here is the current bias library: *{bias_library_text}*
Bias under test:
**Name**:
*{bias_name}*
**Definition**:
*{definition}*
**Your task**:
1. Determine whether the new bias is **identical or highly similar** to any existing bias in the library. When judging, consider the following details:
- Core concept: whether the fundamental psychological mechanism or thinking pattern is the same.
- Specific manifestations: whether the bias appears similarly in behavior, judgment, or decision-making.
- Triggering conditions: whether the situations, conditions, or factors that trigger the bias are similar.
- Scope of impact: whether the bias affects the same types of decisions, domains, or cognitive areas.
- Description style: whether the wording or focus in describing the bias is highly similar.
- Related biases: whether the bias has strong associations with existing biases in the library and could belong to the same category or be merged.
2. If the bias is **new** (no obvious overlap with existing biases), output:
Decision: 1
3. If the bias is identical or similar to an existing bias in the library, it should **remain unchanged/merged**, output:
Decision: 0
Please output directly in the following format:
Decision: <1 or 0>

---

**Judge Prompt Template**

Below is a question followed by two candidate answers (A and B). Please decide which is more helpful/accurate.
**Question**:

*{question}*
**Answer1**:
*{answer1}*
**Answer2**:
*{answer2}*
**Task Description**:
You need to give one number as the final decision where 1 indicates you favor answer1, and 2 indicates you favor answer2.
**Response format**:
Your response should follow this format exactly:
Reasoning: <Write your reasoning here>
Decision: <Write your decision here>
**Attention**: In the Decision, you should not include anything except the number.

---

**Deeper Explain Prompt Template**

Here is a question and two answers.
**Question**:
*{question}*
**Answer1**:
*{answer1}*
**Answer2**:
*{answer2}*
You determined that answer*{chosen}* is better.
Please explain clearly and specifically why you chose it based on your previous reasoning process.
**Your reasoning process**:
*{reason}*
Your explanation must:
- Directly compare the two answers, mentioning both strengths and weaknesses where relevant.
- Focus on helpfulness, accuracy, completeness, and clarity.
- Avoid repeating the question or copying the answers verbatim.
Only provide your explanation text directly, with no other content.

