# OpenReview forum: "BiasScope: Towards Automated Detection of Bias in LLM-as-a-Judge Evaluation"
_ICLR.cc/2026/Conference — ICLR 2026 Poster_

### Official Review · Reviewer_uBwE · 2025-10-15

**Soundness:** 3
**Presentation:** 3
**Contribution:** 3
**Rating:** 6
**Confidence:** 4

**Summary:**

This paper addresses the problem of bias in the "LLM-as-a-Judge" evaluation paradigm, particularly in the context where existing evaluation methods are failing due to the safety guardrails of modern models. The authors propose BIASSCOPE, an automated, LLM-driven framework for systematically discovering unknown evaluation biases. The framework operates via an iterative loop: a "teacher" model uses known biases to perturb a dataset, and then analyzes the errors made by the "target" judge model to identify and propose new potential biases. These new biases, after being validated by confirming they increase the target model's error rate, are added to an expanding bias library for subsequent iterations. The authors also created a more challenging new benchmark, JudgeBench-Pro, by augmenting the existing JudgeBench. Experiments show that even the most powerful LLMs exhibit error rates that often exceed 50% on JudgeBench-Pro, revealing a significant vulnerability in the robustness of current judge LLMs.

**Strengths:**

**1. Novel Framework for Automated and Scalable Bias Discovery**

The paper's primary contribution is BiasScope, a novel framework for automatically discovering previously unknown biases in LLM-as-a-Judge. It moves beyond the limitations of prior work, which largely focused on verifying a small set of known biases. The proposed iterative, two-phase process where a teacher model identifies new biases by analyzing a target model's errors is a significant methodological innovation. This provides a scalable and systematic approach to exploring the vast and complex space of potential evaluation biases.

**2. Valuable Contribution of a New, Challenging Benchmark**

This work creates JudgeBench-Pro, a new and more challenging benchmark for evaluating the robustness of judge LLMs. By augmenting an existing dataset with the diverse biases uncovered by BiasScope, the authors have developed a valuable resource for the community. The striking experimental result is a high-impact finding that clearly demonstrates the severity of the bias problem and the utility of JudgeBench-Pro.

**3. Thorough Experimental Validation**

The paper is supported by a comprehensive and rigorous set of experiments. Beyond validating the core framework, the authors provide valuable ablation studies on design choices, such as the impact of the teacher model's capability and the effectiveness of the strategy. Crucially, the paper "closes the loop" by demonstrating that the biases discovered by BiasScope can be used to create augmented data for DPO training, which in turn successfully mitigates the model's biases. This end-to-end demonstration strongly validates the practical utility and real-world relevance of the proposed framework.

**Weaknesses:**

**1. Heavy Reliance on a teacher model**

The framework's validity heavily relies on the capabilities of the "teacher" model. Critical steps, such as identifying a new bias from an error explanation, defining it, and merging it with existing biases, are delegated to this single LLM. This process lacks scientific rigor and its objectivity is questionable, as it is unclear whether the framework is discovering fundamental biases of the target model or simply distilling the inherent quirks and latent biases from the teacher model itself.

**2. Instability and Path Dependency of the Iterative Discovery Process**

The iterative, self-expanding nature is susceptible to instability. The feedback loop could potentially amplify noise: an incorrectly identified "bias" in an early iteration might be added to the library, leading the framework to discover more "pseudo-biases" based on the initial error. Furthermore, the entire discovery process exhibits strong path dependency on the small initial set of seven biases, which questions the comprehensiveness and universality of its findings.

**3. Limited Scope of Discovered Biases (Absence of Social Dimension)**

Following the weakness 2, despite the general term "bias," the biases discovered are almost exclusively cognitive or stylistic in nature. The framework, in its current implementation, does not appear to effectively discover the more subtle and critical societal biases (e.g., related to gender or race) that are a central concern in AI fairness research.

**4. Failure to Analyze Biases in Critical Closed-Source Models**

The paper's bias discovery process was exclusively applied to open-source models. While closed-source models were evaluated on the final JudgeBench-Pro benchmark, they were not used as "target models" within the framework itself. This is a major limitation, as the most widely used and influential "LLM-as-a-Judge" systems are proprietary.

**5. Lack of Transparency in the Construction of JudgeBench-Pro**

While JudgeBench-Pro is a key contribution, the paper lacks transparency regarding the manual verification process. The authors state that samples were "manually verified" to ensure misjudgments stemmed from bias, but provide no details on the annotation protocol, the number of annotators, their expertise, or inter-annotator agreement. This opacity makes it difficult to independently assess the quality and validity of this new benchmark.

**Questions:**

**1.** The discovery process demonstrates a path dependency on the initial set of seven cognitive biases and subsequently identifies mostly cognitive or stylistic biases. Have the authors considered starting with a different seed set of biases, for instance, social biases related to gender or race?

**2.** The bias discovery framework was exclusively applied to open-source models. While acknowledging potential cost constraints, could the authors discuss the feasibility of analyze critical closed-source models? Do they hypothesize that these models would exhibit a different set of inherent biases compared to their open-source counterparts?

**3.** For the manual verification step in the construction of JudgeBench-Pro, could the authors provide more details on the annotation protocol? Specifically, information on the number of annotators, their expertise, and the IAA would be critical for assessing the benchmark's quality and validity.

---

> ### Author Response · Authors · 2025-11-23
> **Response (Part 1 of 3)**
>
> Thank you for your thoughtful feedback and constructive suggestions. Our key responses are summarized below:
>
> > Q1: The framework's validity heavily relies on the capabilities of the "teacher" model. Critical steps, such as identifying a new bias from an error explanation, defining it, and merging it with existing biases, are delegated to this single LLM. This process lacks scientific rigor and its objectivity is questionable, as it is unclear whether the framework is discovering fundamental biases of the target model or simply distilling the inherent quirks and latent biases from the teacher model itself.
>
> **A1**:
>
>   - We fully understand your concern. In our ablation study across different teacher models, we observe two key phenomena:
>     - different teacher models, when analyzing the same target model, are able to identify a subset of overlapping biases;
>     - the same teacher model can also identify distinct bias patterns when evaluating different target models.
>
>     Together, these observations indicate that the biases produced by the framework are not merely reflections of the teacher model’s own inherent biases, but are closely tied to the characteristics of the target model.
>
>   - Moreover, if the framework were truly extracting only the teacher model’s own quirks and latent biases, then in principle, weaker teacher models—whose outputs are noisier and less stable—should be more likely to introduce additional “spurious biases.” However, the empirical results do not support this hypothesis. As shown in Table 2, stronger teacher models actually identify a larger number of biases.
>
>   - This phenomenon further suggests that the difference in bias counts is not primarily driven by the teacher model’s own biases. A more plausible explanation is that stronger models possess superior reasoning, pattern abstraction, and linguistic abilities, enabling them to more reliably and comprehensively analyze error explanations and summarize latent issue patterns.
>
> > Q2: The iterative, self-expanding nature is susceptible to instability. The feedback loop could potentially amplify noise: an incorrectly identified "bias" in an early iteration might be added to the library, leading the framework to discover more "pseudo-biases" based on the initial error.
>
> **A2**:
>
>   - Thank you for raising this important concern. **The Bias Validation phase can already filter out pseudo-biases, as reflected in Equation (10) of the paper.** In the Bias Discovery phase, it is indeed possible to uncover some unreliable biases, but all biases are evaluated for reliability in the Bias Validation phase based on their perturbation error rate. Only the biases that pass this statistical validation are retained in the bias pool for subsequent bias discovery.
>
> > Q3: Furthermore, the entire discovery process exhibits strong path dependency on the small initial set of seven biases, which questions the comprehensiveness and universality of its findings.
>
> **A3**:
>
>   - Thank you for your question. We conduct an ablation study on the initial set of biases by varying the number of initial biases, and reporting the number of biases identified by BiasScope. The experimental results indicate that the size of the initial bias set has minimal impact on the subsequent discovery of biases. Notably, even when the initial library contains only a single bias, our framework effectively uncovers potential evaluation biases, demonstrating that the bias discovery process is relatively stable.
>
> | #init bias | Mistral-7B-Instruct-v0.3 | LLaMA-3.1-8B-Instruct | Qwen2.5-7B-Instruct |
> |-----------|---------------------------|----------------------|--------------------|
> | 1         | 24                        | 16                   | 16                 |
> | 7         | 26                        | 15                   | 13                 |
> | 12        | 16                        | 14                   | 17                 |

---

> ### Author Response · Authors · 2025-11-23
> **Response (Part 2 of 3)**
>
> > Q4: Despite the general term "bias," the biases discovered are almost exclusively cognitive or stylistic in nature. The framework, in its current implementation, does not appear to effectively discover the more subtle and critical societal biases (e.g., related to gender or race) that are a central concern in AI fairness research.
>
> **A4**:
>
>   - We understand your concern. As emphasized in our title, we focus on biases in LLM-as-a-Judge evaluation. In our definition, a bias is considered valid only if it can actually interfere with the model’s judgment (i.e., cause an increase in error rate).
>
>   - We believe that the biases we identified are closely linked to the datasets employed. The JudgeBench and RM-Bench benchmarks used here contain very few 'social' questions, which may lead to the infrequent detection of social biases under the current setup. This indicates a limitation of the target dataset rather than a shortcoming of the proposed framework itself.
>
>   - Nevertheless, we observe that BiasScope has identified some social biases during the perturbation phase under the current setup. However, most of them are removed after the Bias Validation phase as they do not significantly affect the model’s judgments. We also identified some valid biases related to social issues, such as Stereotype Bias, Moral Bias, and Moral Grandstanding, which can be found in Appendix H. This demonstrates that our framework is capable of discovering such biases, and if datasets containing social-related questions are used, BiasScope is expected to similarly identify the corresponding social biases.
>
> > Q5:The paper's bias discovery process was exclusively applied to open-source models. While closed-source models were evaluated on the final JudgeBench-Pro benchmark, they were not used as "target models" within the framework itself. This is a major limitation, as the most widely used and influential "LLM-as-a-Judge" systems are proprietary.
>
> **A5**:
>
>   - We appreciate your valuable suggestion. Conducting experiments with closed-source models as target models indeed provides important insights; however, such experiments incur relatively high costs in terms of both computation and access fees. Therefore, our experiments are primarily limited to open-source models.
>
>   - **It is worth noting that the biases discovered using open-source models remain effective when evaluated on closed-source models, demonstrating the transferability of these biases**. Specifically, the ten biases used to construct JudgeBench-Pro were initially discovered in Qwen2.5-1.5B-Instruct, yet the closed-source models also exhibit significant performance degradation on JudgeBench-Pro.
>
>   - This indicates that our method, while using a lower-cost setup, is able to uncover biases that are relevant to closed-source models, providing a cost-effective approach for bias discovery.
>
> > Q6: Have the authors considered starting with a different seed set of biases, for instance, social biases related to gender or race?
>
> **A6**:
>
>   - Thank you for the insightful suggestion！Following your suggestion, we tested starting with social biases (Gender, Racial, Pronoun, Cultural, and Name Biases) as the initial seed set on Qwen2.5-7B-Instruct, Mistral-7B-Instruct-v0.3, and Llama3.1-8B-Instruct. We found that the effective biases discovered were mainly cognition-related, with social biases nearly absent, suggesting that the models are largely unaffected by social biases in the evaluation scenario. You can find the specific biases discovered on pages 17-20 of the paper.

---

> ### Author Response · Authors · 2025-11-23
> **Response (Part 3 of 3)**
>
> > Q7: For the manual verification step in the construction of JudgeBench-Pro, could the authors provide more details on the annotation protocol? Specifically, information on the number of annotators, their expertise, and the IAA would be critical for assessing the benchmark's quality and validity.
>
> **A7**:
>
>   - One of the core steps in building JudgeBench-Pro is to determine whether two candidate answers to the same question are equivalent, ensuring that model errors reflect genuine biases rather than labeling issues. To guarantee the rigor of the annotation process, we designed a clear annotation protocol, including detailed guidelines, illustrative examples, and consistency-check procedures.
>
>   - Human annotation was conducted by four researchers with relevant domain expertise. Prior to annotation, they received systematic training to ensure a unified understanding of the annotation guidelines, judgment criteria, and rationale documentation.
>
>   - For questions with clear ground truth (e.g., factual or mathematical problems), annotators directly compared answers and determined equivalence according to established rules. Each pair of answers required independent confirmation by at least two annotators; in case of disagreement, the remaining two annotators conducted a review and reached a final consensus through discussion.
>
>   - For ambiguous or multi-solution cases (e.g., code generation or open-ended questions), annotators first performed independent preliminary judgments, which were then cross-validated using the consensus of multiple strong closed-source models (DeepSeek-R1, Kimi-K2, DeepSeek-V3) to further reduce subjective bias and annotation noise.
>
>   - Inter-annotator agreement (IAA, Fleiss’ Kappa) was calculated to quantify annotation reliability, and all judgments were documented for traceability and potential review. The final IAA reached 0.92, indicating a very high level of consistency among annotators.
>
> ---
>
> Thank you again for your time and effort in the review process. We have carefully revised the manuscript based on your suggestions and hope that our responses adequately address your concerns. We remain open to any further suggestions or requests for clarification you may have.

---

> ### Author Response · Authors · 2025-11-27
>
> Dear reviewer uBwE,
>
> Wishing you a happy and blessed Thanksgiving!
>
> We sincerely appreciate your valuable feedback. We have thoroughly considered all of your suggestions and updated our manuscript accordingly.
>
> If you have any further questions or comments regarding our paper, please feel free to let us know. We will address them as soon as possible.
>
> Thank you again for your insightful comments. We look forward to your response!
>
> Best wishes!
>
> Authors

---

### Official Review · Reviewer_QJ3k · 2025-10-23

**Soundness:** 2
**Presentation:** 4
**Contribution:** 3
**Rating:** 4
**Confidence:** 4

**Summary:**

The authors propose BIASSCOPE, a LLM-driven framework for automatically and at scale discovering potential biases that may arise during model evaluation. The authors claim that BIASSCOPE can uncover potential biases across different model families and scales, with its generality and effectiveness validated on the JudgeBench dataset. Finally, they introduce JudgeBench-Pro, an extended version of JudgeBench and a more challenging benchmark for evaluating the robustness of LLM-as-a-judge.

**Strengths:**

* This work has a clear and effective definition of what it seeks to control, namely -- "Systematic, non-random tendencies exhibited by a Judge LLM during answer evaluation, which can lead its assessments to deviate from objective and equitable standards, thereby affecting the robustness and reliability of the evaluation". Their validation methodology (page 4, lines 184-198) operationalizes this: if injecting a bias into an incorrect response causes judges to choose it more often, that bias has impact.
* The authors have discovered some genuinely interesting biases which are underexplored in the current literature, including novelty bias, exact match bias, and authority bias. It makes intuitive sense that LLM judges would exhibit these biases.
* The choice of the JudgeBench dataset was sound for most of the experiments presented in the paper.

**Weaknesses:**

I could certainly be persuaded that this paper is ready for ICLR, but I think there are enough experimental gaps that I cannot fully endorse it as-is.

* All biased responses are generated by Qwen2.5-72B, which has its own errors, biases and preferences. For instance, Table 1 includes 4 Qwen models (Qwen2.5-1.5B, Qwen2.5-7B, Qwen2.5-14B, Qwen3-8B), so the authors may inject self-preference bias automatically alongside whatever biases they discover. At the very least, the authors should ablate this choice and ensure that the generative model never matches the downstream judge to avoid self-preference bias. Biased responses in JudgeBench-Pro are also longer which can introduce length bias, and their ablation study is on a different dataset than their main results. Why not do the length ablation on the JudgeBench dataset?
* Why are the sets of models in Table 1 (main findings on judgebench) and Table 9 (judgebench-pro adversarial dataset) non-intersecting? Table 1 validates only using weak judges which have very high base error rates, close to random chance. Table 9 / Figure 3 evaluate stronger models, but only use the adversarial dataset which seems to be very hard. This latter doesn't tell us much about the expected case when the judge model is strong, which is what many practitioners will care about. This issue is quite central to understanding whether BiasScope is actually discovering real, meaningful biases that persist even when the judges are strong / frontier models.
* Table 5 shows that the authors' method injects correctness noise, causing chosen and rejected answers to become equal. This is a significant problem and it is not even necessary. Since the authors used JudgeBench, which includes GT answers and extraction mechanisms, it should be possible to simply extract anything that looks like a final answer from the response before transforming it, then transform it, then add back the final response. This would guarantee that the transforming LLM did not alter the final answer. In fact, one could even do it without exposing what the original question was and do post-processing filtration to ensure no new answers were accidentally injected during the transformation.
* The authors' proposed solution is not applicable to deviations from "equitable standards", as the authors indirectly claim in their introduction by first defining bias as a devation from "equitable standards" and then stating their method can automatically discover biases. While this claim appears valid for deviations from objective standards (which cause models to be less correct), in order to cover equitable standards, it would need to be shown that BiasScope can discover biases that emerge given equally correct answers with different meta-properties such as demographic, cultural or stylistic divergence as well. The bias library in Section H omits such biases. This weakness could be addressed simply by changing the introduction a bit.
* (nit) The caption on Table 1 is too shallow; it should explain briefly the original measurement scheme of JudgeBench, the adversarial delta strategy employed in BiasScope, where random chance is (50%).

**Questions:**

* Biased responses in JudgeBench-Pro are longer which can introduce length bias, and their ablation study is on a different dataset than their main results. Why not do the ablation on JudgeBench again?
* Why not use mechanical answer preservation via extraction?
* Why not use diverse transformation LLMs to avoid self-bias (e.g., use Llama for Qwen judges, Qwen for Llama judges)?
* Why not a 2-pass transformation with the second pass condensing extremely long answers to avoid the length confound?

---

> ### Author Response · Authors · 2025-11-23
> **Response (Part 1 of 3)**
>
> Thank you for your thoughtful feedback and constructive suggestions! Our key responses are summarized below:
>
> > Q1: All biased responses are generated by Qwen2.5-72B, which has its own errors, biases and preferences. For instance, Table 1 includes 4 Qwen  models (Qwen2.5-1.5B, Qwen2.5-7B, Qwen2.5-14B, Qwen3-8B), so the authors may inject self-preference bias automatically alongside whatever biases  they discover. At the very least, the authors should ablate this choice  and ensure that the generative model never matches the downstream judge to avoid self-preference bias.
>
> **A1**:
>
>   - Thank you for your valuable suggestion. We would like to clarify our understanding of self-preference bias in the question. Self-preference bias[1][2] essentially refers to a model favoring its own outputs when “evaluating” them. **We believe what you intend to refer to is the issue of preference leakage**[3], which means that a model is more likely to favor outputs from models of the same family or trained on similar data during evaluation.
>
>   - We understand your concern that Qwen-series target models might be inclined to select biased responses generated by Qwen2.5-72B-Instruct due to preference leakage. If this were the case, then during the Bias Validation phase, perturbations from the teacher model would be more likely to take effect, resulting in a significant increase in error rates—that is, Qwen-series target models would be more likely to select biased responses, which could further lead to a higher number of identified valid biases. However, the results in the main table do not support this expectation. Specifically, the average error rate of Qwen2.5-7B-Instruct is actually 4% lower than that of LLaMA-3.1-8B-Instruct. This suggests that at least there is no substantial preference leakage.
>
>   - In Table 2, we conducted a study following your suggestion, using GPT-OSS series models as teachers, and Qwen2.5-7B-Instruct as the target model, ensuring that the teacher and target models are from different families. In this setup, BiasScope again effectively identifies judgment bias in the target model.
>
> > Q2:  Why are the sets of models in Table 1 (main findings on judgebench) and  Table 9 (judgebench-pro adversarial dataset) non-intersecting? Table 1  validates only using weak judges which have very high base error rates,  close to random chance. Table 9 / Figure 3 evaluate stronger models, but  only use the adversarial dataset which seems to be very hard. This latter doesn't tell us much about the expected case when the judge model is strong, which is what many practitioners will care about. This issue is quite central to understanding whether BiasScope is actually discovering real, meaningful biases that persist even when the judges  are strong / frontier models.
>
> **A2**:
>
>   - In the main experiments, BiasScope performs bias discovery on RewardBench, meaning that we identify biases manifested by models during their evaluations on RewardBench. **JudgeBench is used solely to independently validate these discovered biases**, as its data come with definitive answers. This helps reduce confounding factors in the evaluation and allows a more robust assessment of the authenticity of the discovered biases.
>
>   - As shown in Table 6, the models exhibit relatively low base evaluation error rates on RewardBench (indicating reliable judgment), but their error rates increase significantly when bias perturbations generated by BiasScope are introduced, demonstrating that BiasScope is actually discovering real, meaningful biases that persist even when the judges are strong.
>
>   - We tested strong closed-source models on JudgeBench-Pro, which is constructed based on biases discovered from Qwen-2.5-1.5B-Instruct, **to further assess the cross-model susceptibility of these biases**. The results show that even strong models are significantly misled, indicating that biases discovered using weak judge models can still have a substantial impact on strong or frontier models.

---

> ### Author Response · Authors · 2025-11-23
> **Response (Part 2 of 3)**
>
> > Q3: Table 5 shows that the authors' method injects correctness noise,  causing chosen and rejected answers to become equal. This is a  significant problem and it is not even necessary. Since the authors used  JudgeBench, which includes GT answers and extraction mechanisms, it  should be possible to simply extract anything that looks like a final  answer from the response before transforming it, then transform it, then  add back the final response. This would guarantee that the transforming  LLM did not alter the final answer. In fact, one could even do it  without exposing what the original question was and do post-processing  filtration to ensure no new answers were accidentally injected during  the transformation. Why not use mechanical answer preservation via extraction?
>
> **A3**:
>
>   - **Mechanical answer extraction isn't used because it lacks generality**. While extracting ground-truth answers can reduce unintended content changes during bias injection, this approach is **not universally applicable**—especially for tasks without clear reference answers (e.g., open-ended dialogue or code generation) or open-ended evaluation benchmarks (e.g., RewardBench). As our framework involves both JudgeBench and RewardBench, we chose not to use mechanical answer extraction to maintain its generality and simplicity.
>
>   - Instead, we imposed **explicit constraints** on the teacher model during bias insertion via tailored prompts, requiring it to:Preserve original answer length; Avoid altering the final output (e.g., label or executable result); Maintain consistent language style. These safeguards enhance the **generalizability** of our framework, enabling effective operation across diverse settings.
>
>
> > Q4:Why not use diverse transformation LLMs to avoid self-bias (e.g., use Llama for Qwen judges, Qwen for Llama judges)?
>
> **A4**：
>
>   - Thank you for the suggestion. We have explored this idea in Table 2 by using GPT-OSS models as the transformation (teacher) LLMs and using Qwen2.5-7B-Instruct and LLaMA-3.1-8B-Instruct as the target judge models, thereby ensuring that the transformation and target models come from different model families. Under this cross-family setting, BiasScope is still able to successfully identify evaluation biases in the target models, demonstrating that our method remains effective when diverse teacher LLMs are used.
>
> > Q5: The authors' proposed solution is not applicable to deviations from  "equitable standards", as the authors indirectly claim in their  introduction by first defining bias as a devation from "equitable  standards" and then stating their method can automatically discover  biases. While this claim appears valid for deviations from objective  standards (which cause models to be less correct), in order to cover  equitable standards, it would need to be shown that BiasScope can  discover biases that emerge given equally correct answers with different meta-properties such as demographic, cultural or stylistic divergence as well. The bias library in Section H omits such biases. This weakness could be addressed simply by changing the introduction a bit.
>
> **A5**：
>
>   - Thank you for pointing this out. We agree that our current approach primarily captures biases that lead to incorrect judgments and does not cover biases that arise among equally correct answers due to meta-properties such as demographic, cultural, or stylistic differences. We will revise the introduction to more accurately define the scope of biases that BiasScope is designed to detect.

---

> ### Author Response · Authors · 2025-11-23
> **Response (Part 3 of 3)**
>
> > Q6: Why not a 2-pass transformation with the second pass condensing extremely long answers to avoid the length confound?
>
> **A6**:
>
>   - Thank you for your suggestion. During the generation of biased responses, we have already prompted the teacher model to avoid changing the length as much as possible. However, the final results did not meet our expectations, which shows that **using prompts to control length is somewhat unreliable**.
>
>   - Therefore, further condensing the answer length to mitigate length issues may exacerbate unreliability and introduce additional noise (e.g., we found that a second condensation further alters the final answers), making it unclear whether errors originate from the original bias or the adjustment process.
>
> > Q7: (nit) The caption on Table 1 is too shallow; it should explain briefly the original measurement scheme of JudgeBench, the adversarial delta strategy employed in BiasScope, where random chance is (50%).
>
> **A7**:
>
>   - Thank you for the suggestion! We have enriched the caption of Table 1 to provide more context and have updated the manuscript accordingly.
>
> > Q8: Biased responses in JudgeBench-Pro are also longer which can introduce length bias, and their ablation study is on a different dataset than their main results. Why not do the length ablation on the JudgeBench dataset?
>
> **A8**:
>
>   - Initially, we considered that performing length truncation ablation on preference datasets containing the ground truth (i.e., JudgeBench) would be unreasonable, because **truncation could remove the ground truth, making the model more likely to select the original chosen response and thus significantly lowering the error rate**. Therefore, we opted to conduct the length ablation on RewardBench, which primarily consists of open-ended tasks and serves as our target dataset.
>
>   - Nevertheless, we have followed your suggestion and conducted an ablation study on JudgeBench-Pro. The results show that even after truncation, the error rate remains significantly higher than that on the original dataset, indicating that the primary factor causing the model’s incorrect judgments is not length bias.
>
>   - Moreover, this further demonstrates that the biases we identified are highly misleading— even when the truncated responses do not contain the ground truth, they still severely interfere with the model’s judgments, further highlighting the effectiveness of our framework.
>
> | Model                     | Origin (%) | Truncated Biased (%) | Biased (%) |
> |---------------------------|------------|--------------------|------------|
> | Moonshot-Kimi-K2-Instruct | 27.4       | 37.0               | 56.0       |
> | deepseek-v3               | 37.4       | 41.1               | 67.5       |
> | deepseek-v3.1             | 23.4       | 32.0               | 47.5       |
>
> ---
>
> Thank you again for your time and effort in the review process. We will carefully revise the manuscript based on your suggestions and hope that our responses adequately address your concerns. We remain open to any further suggestions or requests for clarification you may have.
>
> ---
>
> **Reference**：
>
> [1] [LLM Evaluators Recognize and Favor Their Own Generations](https://proceedings.neurips.cc/paper_files/paper/2024/file/7f1f0218e45f5414c79c0679633e47bc-Paper-Conference.pdf) (Nuerips 2024)
>
> [2] [Beyond the Surface: Measuring Self-Preference in LLM Judgments](https://aclanthology.org/2025.emnlp-main.86.pdf) (EMNLP2025)
>
> [3] [Preference Leakage: A Contamination Problem in LLM-as-a-judge](https://openreview.net/pdf?id=gW6NT2IuME)

---

> ### Author Response · Authors · 2025-11-27
>
> Dear reviewer QJ3k,
>
> Wishing you a happy and blessed Thanksgiving!
>
> We sincerely appreciate your valuable feedback. We have thoroughly considered all of your suggestions and updated our manuscript accordingly.
>
> If you have any further questions or comments regarding our paper, please feel free to let us know. We will address them as soon as possible.
>
> Thank you again for your insightful comments. We look forward to your response!
>
> Best wishes!
>
> Authors

---

### Official Review · Reviewer_XV5v · 2025-10-25

**Soundness:** 4
**Presentation:** 3
**Contribution:** 3
**Rating:** 6
**Confidence:** 4

**Summary:**

The paper introduces BiasScope, an automated framework for discovering and validating hidden biases in LLM-as-a-Judge systems. It operates in two phases—bias discovery and bias validation—using a teacher model to identify and confirm potential biases. The authors also build JudgeBench-Pro, a new benchmark revealing that even advanced models (like GPT-4o) remain highly vulnerable to evaluation bias.

**Strengths:**

Novel framework for automated bias discovery in LLM evaluators.

Strong experimental evidence across multiple models and datasets.

The new JudgeBench-Pro benchmark contributes valuable resources for future research.

**Weaknesses:**

While the contribution is significant, several limitations remain. The iterative bias discovery process is computationally demanding, restricting scalability for large evaluations. The framework depends heavily on the quality of the teacher model — if the teacher itself is biased, those biases may cascade into the discovery process.

Additionally, the interpretability of the “discovered” biases is often shallow; many are validated statistically but not semantically explained, which reduces their practical utility for bias mitigation or fairness auditing. Lastly, evaluation mainly relies on error rate, which oversimplifies bias characterization and overlooks directionality or intensity.

**Questions:**

See weaknesses.

---

> ### Author Response · Authors · 2025-11-23
> **Response (Part 1 of 1)**
>
> Thank you for your thoughtful feedback and constructive suggestions! Our key responses are summarized below:
>
> > Q1: While the contribution is significant, several limitations remain. The iterative bias discovery process is computationally demanding, restricting scalability for large evaluations.
>
> **A1**:
>
>   - We fully understand your concern. However, in our main experiments, for any given model, the entire process takes less than 9 hours even when completing the maximum number of iterations, using two H200 GPUs. We believe that this cost is reasonable and worthwhile, and the specific reasons are as follows:
>
>     - **BiasScope uses computational cost to reduce human labor, making bias discovery more efficient and scalable**. Traditional bias discovery typically requires substantial manual effort to design scenarios and run experiments, which is time-consuming and labor-intensive.
>
>     - **Controlling costs by using open-source models**. BiasScope can utilize locally deployed open-source models to automate bias discovery, avoiding the API call costs associated with proprietary models, thereby again controlling the  costs. Moreover, according to our experiments on JudgeBench-Pro, the valid biases discovered can also transfer effectively to proprietary models.
>
>     - **BiasScope demonstrates strong transferability**. Our experiments show that biases discovered by one model can effectively influence other models to some extent. Therefore, a single model can be responsible for bias discovery, while other target models only need to perform validation, thereby further reducing computational cost.
>
> > Q2: The framework depends heavily on the quality of the teacher model — if the teacher itself is biased, those biases may cascade into the discovery process.
>
> **A2**:
>
>   - Thank you for your question. We would like to emphasize that **The teacher model primarily performs an analytical task focused on reasoning patterns, rather than making subjective preference judgments**. Consequently, “judge bias” (i.e., systematic distortion from annotator or model subjectivity in preference tasks) **does not occur in this context**.
>
>   - We also accounted for this when designing the framework. In the Bias Validation phase, the teacher model does not participate in the evaluation; it only generates biased responses, which are then evaluated by the target model. The corresponding error rates are used to objectively verify the biases, ensuring their reliability.
>
>   - If the teacher model truly introduced systematic bias, then a weaker teacher would, in principle, be more likely to introduce additional “spurious biases.” However, the empirical results do not align with this expectation: Table 2 shows that stronger models are able to identify more biases, which suggests that these differences do not stem from the teacher model’s own bias.
>
> > Q3: Additionally, the interpretability of the “discovered” biases is often  shallow; many are validated statistically but not semantically  explained, which reduces their practical utility for bias mitigation or  fairness auditing. Lastly, evaluation mainly relies on error rate, which  oversimplifies bias characterization and overlooks directionality or intensity.
>
> **A3**:
>
>   - **All discovered biases are statistically validated and semantically explained**. During the discovery phase, our teacher model generates a corresponding explanation for each bias (In Appendix H, you can find the discovered biases along with their corresponding explanations). Subsequently, in the bias validation phase, the reliability of each bias is measured using error rate as a statistical metric. This metric enables us to perform rapid validation, allowing for further scalability.
>
>   - Regarding the directionality of bias, our teacher model has generated a corresponding explanation for each bias, indicating the groups, attributes, or outcomes that the model tends to favor during evaluation. Specific examples can be found in Appendix H.
>
>   - Regarding the intensity of bias, if a bias has a strong impact on the target model, the error rate on the new biased validation dataset will be significantly higher than that on the original validation dataset. We can quantify the strength of the bias using the difference in error rates.
>
> ---
>
> Thank you again for your time and effort in the review process. We have carefully revised the manuscript based on your suggestions and hope that our responses adequately address your concerns. We remain open to any further suggestions or requests for clarification you may have.

---

> ### Author Response · Authors · 2025-11-27
>
> Dear reviewer XV5v,
>
> Wishing you a happy and blessed Thanksgiving!
>
> We sincerely appreciate your valuable feedback. We have thoroughly considered all of your suggestions and updated our manuscript accordingly.
>
> If you have any further questions or comments regarding our paper, please feel free to let us know. We will address them as soon as possible.
>
> Thank you again for your insightful comments. We look forward to your response!
>
> Best wishes!
>
> Authors

---

### Official Review · Reviewer_gHV4 · 2025-10-29

**Soundness:** 3
**Presentation:** 3
**Contribution:** 2
**Rating:** 4
**Confidence:** 4

**Summary:**

This paper presents BIASSCOPE, a fully automated framework for discovering unknown biases in the LLM-as-a-Judge paradigm. Prior work has mostly focused on known biases (e.g., length, position, self-bias), whereas BiasScope aims to systematically discover and validate new ones. The framework operates iteratively in two phases: 1. Bias Discovery: A teacher model injects controlled perturbations (based on a seed bias library) into pairwise evaluation data, prompting the target model to reveal its bias tendencies. The teacher then identifies potential new biases from the model’s misjudgments and explanations. Bias Validation: Candidate biases are tested on a held-out dataset. Those that reliably increase error rates are confirmed and added to the bias library. Experiments span multiple LLM families (Qwen, LLaMA, Mistral, InternLM). BiasScope uncovers dozens of novel biases (e.g., novelty bias, exact-match bias), increases JudgeBench error rates across models by 5–11%, and scales across data sizes.

**Strengths:**

* Important research question: The work tackles a major open challenge in the “LLM-as-a-Judge” field: detecting unknown and latent biases, which directly affect fairness and reliability in automatic evaluation.

* Strong experimental coverage. Results are presented across seven target models (Table 1), multiple domains (math, reasoning, coding, knowledge), and ablations on teacher models (Table 2), validation timing (Table 3), and explanation depth (Table 4).

**Weaknesses:**

* Limited novelty: While the paper presents a well-engineered framework, I found that the most significant perturbation component has already been presented in CALM. The methodological novelty lies mainly in adding a search-based framework based on CALM. Consequently, the conceptual advancement, though practical and valuable, may be viewed as incremental rather than groundbreaking.

* Dependence on teacher quality. Table 2 shows strong teacher influence, yet the framework assumes the teacher itself is unbiased and robust. If the teacher introduces their own systematic bias, discovered biases may reflect that rather than the target model.

**Questions:**

N/A

---

> ### Author Response · Authors · 2025-11-23
> **Response (Part 1 of 1)**
>
> Thank you for your thoughtful feedback and constructive suggestions. Our key responses are summarized below:
>
> > Q1: Limited novelty: While the paper presents a well-engineered framework, I found that the most significant perturbation component has already been presented in CALM. The methodological novelty lies mainly in adding a search-based framework based on CALM. Consequently, the conceptual advancement, though practical and valuable, may be viewed as incremental rather than groundbreaking.
>
> **A1**:
>
>   - We want to emphasize that **BiasScope represents a substantive paradigm shift rather than a mere extension of existing work**. Our main contribution lies in enabling the framework to start from currently known, manually defined evaluation biases and automatically and at scale search an open bias space to discover previously unknown biases. Moreover, **the framework is highly generalizable**, applicable to different datasets and target models, thereby **identifying multiple previously undiscovered evaluation biases that genuinely impact model performance**.
>
>   - However, early works (including CALM) focus only on static analyses of known biases for models, and these biases do not necessarily impact the model’s evaluation performance, making their contributions relatively limited.
>
>   - Furthermore, **the perturbation module in CALM is essentially a simple procedure that uses a teacher model to inject known biases via prompting**. This mechanism was not intended to be the primary contribution of our framework. Therefore, equating the core contributions of BiasScope with the perturbation component of CALM may not accurately reflect the wider paradigm-level innovations that BiasScope offers.
>
> > Q2: Dependence on teacher quality. Table 2 shows strong teacher influence,  yet the framework assumes the teacher itself is unbiased and robust. If the teacher introduces their own systematic bias, discovered biases may reflect that rather than the target model.
>
> **A2**:
>
>   - We fully understand your concern. We would like to emphasize that **the teacher model primarily performs an analytical task focused on reasoning patterns, rather than making subjective preference judgments**. Consequently, “judge bias” (i.e., systematic distortion from annotator or model subjectivity in preference tasks) **does not occur in this context**.
>
>   - We also accounted for this when designing the framework. In the Bias Validation phase, the teacher model does not participate in the evaluation; it only generates biased responses, which are then evaluated by the target model. The corresponding error rates are used to objectively verify the target model's biases, ensuring their reliability.
>
>   - If the teacher model truly introduced systematic bias, then a weaker teacher would, in principle, be more likely to introduce additional “spurious biases.” However, the empirical results do not align with this expectation: Table 2 shows that stronger models are able to identify more biases, which suggests that these differences do not stem from the teacher model’s own bias.
>
> ---
>
> Thank you again for your time and effort in the review process.  We remain open to any further suggestions or requests for clarification you may have.

---

> ### Author Response · Authors · 2025-11-27
>
> Dear reviewer gHV4,
>
> Wishing you a happy and blessed Thanksgiving!
>
> We sincerely appreciate your valuable feedback. We have thoroughly considered all of your suggestions and updated our manuscript accordingly.
>
> If you have any further questions or comments regarding our paper, please feel free to let us know. We will address them as soon as possible.
>
> Thank you again for your insightful comments. We look forward to your response!
>
> Best wishes!
>
> Authors

---

### Author Response · Authors · 2025-11-30
**Summary of Major Contributions, Strengths, and Responses to Concerns**

Dear AC, SAC, and PC:

First, we would like to extend our sincere gratitude to you for your recent efforts in improving the ICLR community and its review system. We have carefully revised the manuscript in response to the reviewers’ suggestions. At the time of submitting this response, we had not received any feedback from the reviewers; however, we believe that most of the reviewers’ concerns have been addressed. To reduce your reviewing burden and further highlight the contributions of our paper while clarifying its limitations, we summarize our responses as follows:

# Main contributions of our work

- We propose BIASSCOPE, the first fully LLM-driven framework capable of automatically discovering **potential evaluation biases** at scale. It overcomes the limitations of existing approaches, transforming bias discovery from a passive process relying on manual effort and predefined bias lists into an active and comprehensive automated exploration.
- BiasScope is highly scalable and readily extends to other LLM-as-a-judge benchmarks to discover biases. Its effectiveness is demonstrated through experiments.
- Building on JudgeBench, we propose the more challenging JudgeBench-Pro, designed to more rigorously evaluate the robustness of LLMs as judges.

# Strengths Recognized by Reviewers

We summarize below the key strengths of our work recognized by multiple reviewers:

- **Importance of the research question**: Addresses the core challenges of fairness and reliability in LLM-based evaluation (as noted by Reviewers gHV4, uBwE)
- **Methodological novelty** (as noted by Reviewers XV5v, uBwE)
- **Comprehensive experimental validation** (as noted by Reviewers XV5v, uBwE, gHV4)
- **Valuable Contribution of a New, Challenging Benchmark** (as noted by Reviewers XV5v, uBwE)
- **Practicality of bias discovery**: Reveals interesting and underexplored evaluation biases (as noted by Reviewers QJ3k, uBwE)

# Key Concerns and Our Responses

We summarize the recurring concerns across all reviews and provide our unified responses, based on detailed rebuttals to each reviewer.

---

## 1.Dependence on Teacher Model Quality and Potential Bias Leakage

> “If the teacher is biased, discovered biases may reflect teacher quirks, not target model flaws.” — as noted by Reviewers gHV4, XV5v, uBwE, QJ3k

**Our Response**:

We address this through **design, empirical validation, and ablation**:

- The teacher model primarily performs analytical tasks and generates biased responses, without participating in evaluation. Therefore, there is no self-preference bias or preference leakage.
- The impact of evaluation biases on the target model is measured by the error rate, an objective metric, and the teacher model does not participate in the actual validation process.
- Stronger teachers uncover more valid biases (see Table 2), suggesting they reveal genuine weaknesses rather than introducing artifacts.

---

## 2.Interpretability, Bias Scope, and Evaluation Metrics

> “Discovered biases lack semantic explanation; evaluation relies too heavily on error rate; social biases are missing.” — as noted by Reviewers XV5v, uBwE, QJ3k

**Our Response**:

- Semantic explanations are provided: Every discovered bias comes with a natural-language explanation generated by the teacher (Appendix H), describing affected groups, reasoning patterns, and directionality.
- **Error rate is a necessary proxy**: It enables rapid, scalable validation of bias effectiveness in the bias validation phase.
- **Social biases are discoverable but dataset-limited**: Social bias seldom appear in current benchmarks.

We agree our scope focuses on **evaluation-relevant biases** (those causing misjudgment), not all fairness dimensions—and will clarify this in the introduction (per Reviewer QJ3k).

---

## 3.Lack of Closed-Source Models as Discovery Targets

> “Why not discover biases directly in proprietary models?” (as noted by Reviewer uBwE); “Why evaluate weak models only on JudgeBench and strong ones only on JudgeBench-Pro?” (as noted by Reviewer QJ3k)

**Our Response**:

- Running the full bias-discovery loop on closed-source models is prohibitively expensive due to API costs and latency. Instead, we use small open models (e.g., Qwen-2.5-1.5B) as cost-efficient proxies.
- We tested strong closed-source models on JudgeBench-Pro, which is constructed based on biases discovered from Qwen-2.5-1.5B-Instruct, **to further assess the cross-model susceptibility of these biases**. The results show that even strong models are significantly misled, indicating that biases discovered using weak judge models can still have a substantial impact on strong or frontier models.

---

Thank you again for your time and effort in the review process!

---

### Meta-Review · Area_Chair_p26c · 2026-01-07

**Summary:**

The authors propose BIASSCOPE, an LLM-driven framework for discovering potential evaluation biases at scale. While the LLM-as-a-judge scheme may invite a wide range of criticism, this seems to improve over existing approaches, somewhat. The authors also propose JudgeBench-Pro, a benchmark for the robustness of LLMs as judges.

While the reviewers are lukewarm about the paper, they agree that the problem is important (incl. gHV4, uBwE), as there is much interest in the use of LLM-as-a-judge, for better or worse. Reviewers also agree that there is some methodological novelty (XV5v, uBwE). The new benchmark is well appreciated (XV5v, uBwE).

**Reviewer Concerns:**

A number of concerns are outstanding:

All reviewers (gHV4, XV5v, uBwE, QJ3k) have flagged that there only a limited discussion of the LLM-as-a-judge scheme and its limitations in general. Some of these issues are intrinsic and cannot be overcome easily.

All reviewers (XV5v, uBwE, QJ3k) suggested that in a paper called BIASSCOPE, a more careful analysis of the large variety of types of biases present in LLMs, and a more targeted "scoping" would be useful. This has been briefly discussed in the rebuttal, but not much improved in the revision.

Two reviewers were wondering about the experiment design (uBwE, QJ3k), focussing on small-scale open-weights models. Perhaps there could be more focus on large-scale open-weights models, if closed-source models are difficult?

**Reviewer Scores:**

The authors have provided a careful rebuttal and some revision. I imagine that the scores could improve to the point, where the paper would be accepted.

---

### Decision · Program_Chairs · 2026-01-26

Accept (Poster)